# KSHV transactivator-derived small peptide traps coactivators to attenuate MYC and inhibits leukemia and lymphoma cell growth

Michiko Shimoda [1,2✉], Yuanzhi Lyu[1], Kang-Hsin Wang[1], Ashish Kumar[1], Hiroki Miura[1], Joshua F. Meckler[2,3], Ryan R. Davis [4], Chanikarn Chantarasrivong[5], Chie Izumiya[1], Clifford G. Tepper [2,6], Ken-ichi Nakajima[1], Joseph Tuscano[2,3], Gustavo Barisone[2,3] & Yoshihiro Izumiya [1,2,6✉]

In herpesvirus replicating cells, host cell gene transcription is frequently down-regulated because important transcriptional apparatuses are appropriated by viral transcription factors. Here, we show a small peptide derived from the Kaposi's sarcoma-associated herpesvirus transactivator (K-Rta) sequence, which attenuates cellular MYC expression, reduces cell proliferation, and selectively kills cancer cell lines in both tissue culture and a xenograft tumor mouse model. Mechanistically, the peptide functions as a decoy to block the recruitment of coactivator complexes consisting of Nuclear receptor coactivator 2 (NCOA2), p300, and SWI/SNF proteins to the *MYC* promoter in primary effusion lymphoma cells. Thiol(SH)-linked alkylation for the metabolic sequencing of RNA (*SLAM seq*) with target-transcriptional analyses further confirm that the viral peptide directly attenuates *MYC* and MYC-target gene expression. This study thus provides a unique tool to control MYC activation, which may be used as a therapeutic payload to treat MYC-dependent diseases such as cancers and autoimmune diseases.

[1] Department of Dermatology, School of Medicine, University of California Davis (UC Davis), Sacramento, CA, USA. [2] UC Davis Comprehensive Cancer Center, Sacramento, CA, USA. [3] Department of Internal Medicine, School of Medicine, UC Davis, Sacramento, CA, USA. [4] Department of Pathology and Laboratory Medicine, School of Medicine, UC Davis, Sacramento, CA, USA. [5] Lifematics Inc., Osaka, Japan. [6] Department of Biochemistry and Molecular Medicine, School of Medicine, UC Davis, Sacramento, CA, USA. ✉email: mshimoda@ucdavis.edu; yizumiya@ucdavis.edu

Expression of c-Myc (MYC) is tightly regulated in normal cells but becomes dysregulated and often over-expressed in many types of human cancer cells. Numerous studies suggest that deregulation of MYC expression occurs in >70% of cancers overall and contributes to disease progression, metastatic potential, and therapeutic resistance[1]. Mechanistically, increased MYC protein concentration in the nucleus facilitates its interaction with E-box transcription factors, which triggers tumorigenesis by promoting cell proliferation; in contrast, these events should not occur in resting cells with tightly regulated, normal physiological levels of MYC. The ubiquitous nature of MYC deregulation in most types of cancers and its inherent driver function for cell proliferation make MYC a very attractive target for cancer drug development. However, MYC, like other transcription factors, possesses a highly disordered structure, which facilitates interaction with multiple cofactors and DNA[2,3], but hampers development of specific small molecule inhibitors. Nonetheless, several MYC-specific inhibitors have been developed successfully. Omomyc is a MYC-specific inhibitor, which is a miniprotein 90 amino acids in length derived from the MYC basic helix-loop-helix (bHLH) domain that acts as a dominant negative mutant for the MYC dimerization domain[4]. Four mutations in the leucine zipper region still permit Omomyc to dimerize with all MYC family proteins, but effectively prevent the MYC/MAX heterodimers from binding their target promoters thereby inhibiting MYC-dependent gene expression[5]. As expected, Omomyc inhibits the cell growth of multiple cancer cell types in vitro and in vivo[5].

In addition to directly targeting MYC, the epigenetic silencing of MYC gene expression by targeting histone modifying enzymes or binding proteins also reduces MYC expression. Epigenetic targets include histone deacetylase, acetylase, demethylase, methylase, and bromodomain and extra-terminal motif (BET) proteins. Small molecule inhibitors targeting the respective enzyme pockets or acetylated histone binding surfaces have all shown some efficacies against MYC, with BET inhibitors being the most well-studied[6–11]. For example, the small molecule inhibitor JQ1 was one of the first BET bromodomain inhibitors developed and acts as a mimetic to the acetylated histone tail[12]. The mimetic small molecule competes with BET proteins (mainly BRD2 and 4) for binding to the acetylated histone tail, and pharmacological inhibition of BET proteins shows therapeutic activity in a variety of pathologies, particularly in models of cancer and inflammation[11,13–15]. Recent direct target identification with high-resolution transcription studies demonstrated that BRD4 acts as a coactivator of RNA polymerase II (RNAPII)-dependent transcription; consequently, gene transcription is broadly repressed with high-dose BET bromodomain inhibitor treatment. At doses triggering selective effects in leukemia, the BET bromodomain inhibitors deregulate a small set of hypersensitive targets, which include MYC[9]. Perhaps MYC is particularly sensitive to a general transcription inhibitor due to its higher rate of transcription in cancer cells combined with its short mRNA and protein half-life. However, recent clinical trials showed that BET bromodomain inhibitors induce large-scale transcriptional reprogramming, which leads to frequent drug resistance and thus disappointing efficacy[16].

Kaposi's sarcoma-associated herpesvirus (KSHV), also designated as human herpesvirus 8, is one of eight human herpesviruses. KSHV is the causative agent of Kaposi's sarcoma[17,18], two human lympho-proliferative diseases, primary effusion lymphoma (PEL)[19,20] and multicentric Castleman's disease[21,22], and more recently described to be associated with KSHV-inflammatory cytokine syndrome[23,24]. Like all other herpesviruses, the KSHV lifecycle consists of two phases, latency and lytic replication. In latency, the viral genome persists in the host cells as nuclear episomes, and only a few viral genes are actively transcribed[25,26]. Transitioning from latency to lytic replication is initiated by the expression of a single viral transcription factor, K-Rta (ORF50). The expression of viral K-Rta alone possesses strong transactivation function, which is sufficient to trigger the KSHV lytic gene expression cascade from latent KSHV chromatin[27–31]. K-Rta is classified as an immediate-early gene and its coding sequence consists of two exons[30,32]. The K-Rta protein encodes DNA binding and dimerization domains at the N-terminal, and a transactivation domain at the C-terminal region[33].

Similar to other herpesvirus infections, KSHV replication causes cell cycle arrest at late G1 phase[34,35], and also redirects cellular RNAPII and associated enzymes to viral genomes to meet the great demand for robust viral gene expression and replication to be completed within a short period of time[36–38]. KSHV gene transcripts, which include viral long non-coding RNAs, can reach levels of expression accounting for >40% of the total RNA-sequencing (RNA-seq) reads in KSHV-reactivated RNA samples[39], suggesting that KSHV evolved a highly efficient mechanism to hijack the majority of the host transcription machinery during viral replication. Therefore, isolating and manipulating such a specific viral protein functional domain may be used as a competitor to attenuate cellular protein functions.

In this study, we generated a small peptide drug based on the binding interface of K-Rta and the cellular coactivator complex. The small peptide inhibitor blocks recruitment of the SWI/SNF complex from engaging the MYC promoter and results in downregulation of both MYC and MYC-target gene transcription. Consequently, this peptide inhibited PEL cell growth both in vitro and in vivo. The initial characterization of the peptide-mediated gene repression and an example of its therapeutic potential are described in this report.

## Results

**Identification of the KSHV transactivation complex.** We previously reported that when KSHV reactivation begins, RNA Polymerase II (RNAPII) molecules are effectively recruited to viral episomes and form a complex with a viral protein, KSHV replication and transactivation (K-Rta), for viral gene expression[36]. We examined cellular proteins that are localized in the RNAPII and K-Rta protein complex during KSHV reactivation by utilizing rapid immunoprecipitation mass spectrometry of endogenous protein (RIME)[40]. This method is well-suited to comprehensively identify transcription factor complexes on chromatin. We examined interacting protein partners of K-Rta and RNAPII by immunoprecipitation with anti-Flag antibody for K-Rta and anti-RNAPII antibody, respectively, before (K-Rta is absent) and after (K-Rta is present) triggering reactivation (Fig. 1a). Proteins co-precipitated with RNAPII that were further increased in the presence of K-Rta, and also co-precipitated with K-Rta (anti-Flag antibody) were considered to be recruited by K-Rta for viral gene transactivation. Non-reactivated samples served as a negative control for K-Rta RIME experiments. In addition, RIME conducted with non-specific IgG was used to control for non-specific interactions for both uninduced and induced samples. After stringent filtering with criteria described in the methods section, RIME identified a total of 87 K-Rta interacting proteins (Supplementary Data 1); these include several protein complexes functioning in RNA processing, RNA splicing, chromosome organization, DNA repair, and DNA replication. Furthermore, RNAPII RIME showed that 85 of the K-Rta interacting proteins were among the set of RNAPII-interacting proteins (Supplementary Data 1). Fifteen cellular proteins were identified that newly interacting with RNAPII during KSHV

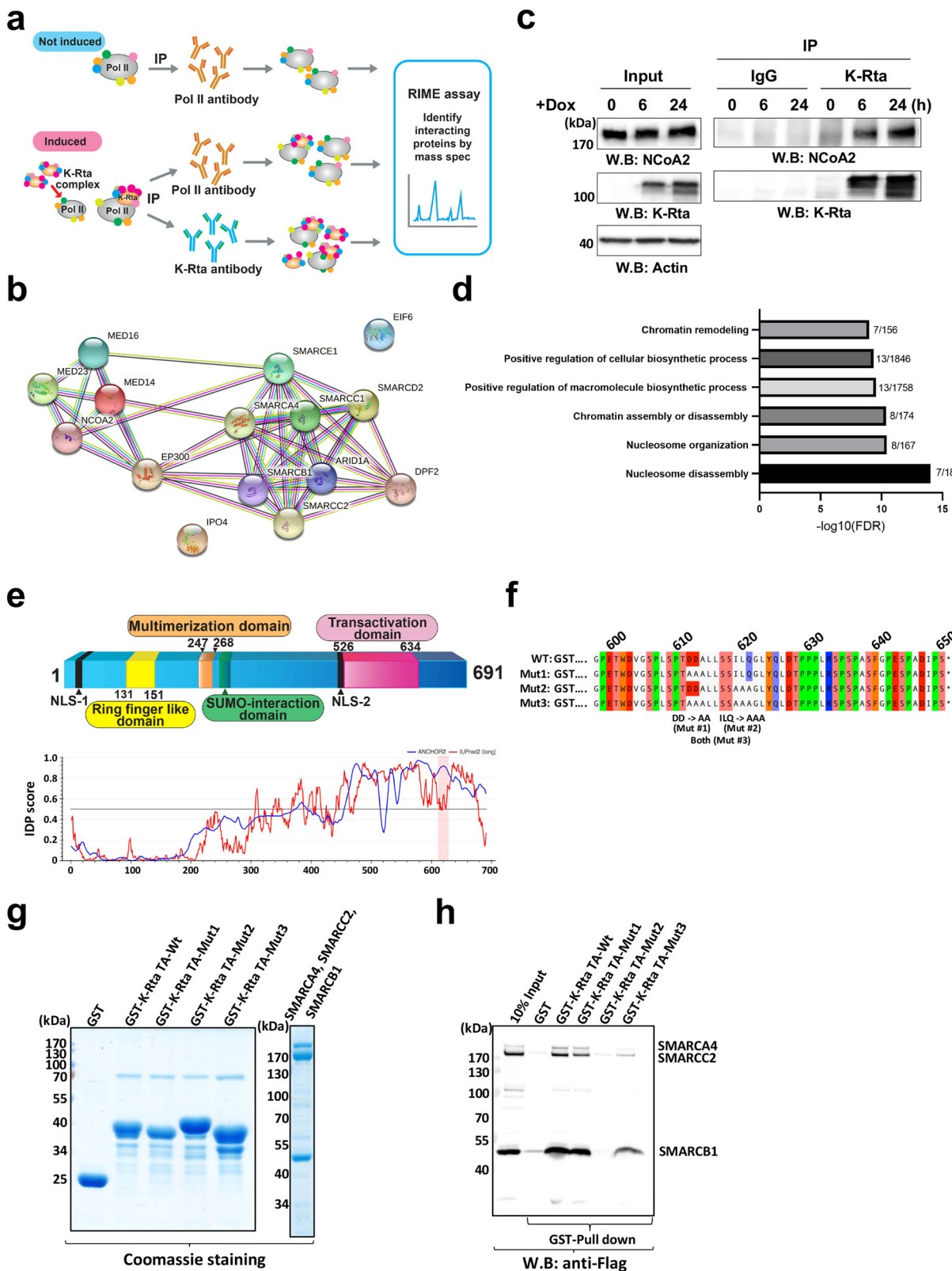

reactivation with >1.5-fold enrichment with $p$ value <0.05 (Fig. 1b). In contrast, proteins that dissociated from RNAPII upon K-Rta induction are known to function in DNA replication and DNA strand elongation (FDR = 5.12E-12) (Supplementary Fig. 1). The 15 RNAPII-interacting proteins for which the interaction is induced by K-Rta expression include 8 SWI/SNF components[41], three mediator complex proteins, p300 histone

acetylase, nuclear receptor coactivator 2 (NCoA2), and a protein that functions in nuclear export. Analysis with STRING, a bioinformatics tool to analyze protein–protein interaction networks, demonstrated these activator complexes could interact as a single large protein complex (Fig. 1b). Overlapping or shared protein interactions with K-Rta also suggested that there is a pre-assembled large coactivator complex. Co-immunoprecipitation

**Fig. 1 KSHV transactivator interacts with cellular coactivator complexes with its intrinsically disordered transactivation domain. a** Scheme for Rapid immunoprecipitation Mass spectrometry of Endogenous protein (RIME) assays. Two antibodies anti- Flag (for K-Rta) and anti-RNAPII (POLR2A) were used to identify cellular proteins that exhibited increased interactions with POLR2A during KSHV reactivation. **b** STRING Interaction map. List of protein names among RNAPII-interacting proteins whose peptide counts were increased >1.5-fold in the presence of K-Rta with *p*-value <0.05 are plotted with STRING. **c** Co-immunoprecipitation. KSHV reactivation was induced by inducing exogenous K-Rta expression from a tetracycline-inducible promoter, and immunoprecipitation was performed with anti-K-Rta antibody. The indicated protein was probed for using the corresponding specific antibody. Total cell lysates were used as input control. **d** Gene ontology of recruited proteins via K-Rta. Protein names were subjected to GO analyses and the six highest protein functions are presented in the table. **e** Schematic diagram of K-Rta protein domains and intrinsically disordered protein (IDP) plot. Previously published K-Rta-function domains are indicated in the diagram. The transactivation domain, with which K-Rta interacts with the coactivator complex, is marked in magenta. The IDP score was calculated with a web-interface (https://iupred2a.elte.hu/) and plotted. The K-Rta transactivation domain is located at an intrinsically disordered region (IDR). **f** K-Rta and mutant transactivation domain sequence. Recombinant GST-protein sequences used for GST-pull down assays are shown. Altered amino acid sequences in the mutant are depicted in the bottom of the panel. **g** Purified GST-K-Rta deletion proteins and SWI/SNF components used for pull-down. Coomassie staining of SDS-PAGE gels are shown in the left panel. Molecular size marker is indicated left side of gel. **h** GST-pull down assays. The indicated purified SWI/SNF proteins were mixed and incubated with GST-K-Rta peptides. Immunoblotting with anti-Flag tag antibody was performed to probe interactions.

with anti-Flag (for K-Rta) antibody showed that interaction between K-Rta and NCoA2 were increased during KSHV reactivation along with an increasing amount of K-Rta protein expression, which confirmed the RIME studies (Fig. 1c). In addition, our RIME studies agreed well with previous reports, which demonstrated that the SWI/SNF complex, mediator complex, and NCoA2 are K-Rta interacting proteins, and that the interaction plays an important role in K-Rta's transactivation function in KSHV-infected cells[37,38]. Functional annotation of KSHV reactivation-induced K-Rta and RNAPII-interacting proteins predicted that these proteins are primarily involved in chromatin disassembly to trigger transcription elongation [false discovery rate (FDR) = 1.00E-14 (Fig. 1d)].

**KSHV K-Rta interacts with coactivators at its transactivation domain.** The KSHV reactivation-induced interaction of coactivators with both RNAPII and K-Rta suggested that K-Rta might physically bring the complex to the RNAPII complex poised on the KSHV genome[42], thereby stimulating KSHV lytic gene expression. To test this hypothesis, we next examined direct-physical interaction between K-Rta and the coactivator complex and mapped the interaction domain(s) in detail. Previous studies with GST-K-Rta fusion proteins showed that K-Rta precipitated SWI/SNF proteins and the mediator complex from cell lysates, and amino acid residues 612–621 within the K-Rta transactivation domain (Fig. 1e) were responsible for the precipitation of the coactivators from cell lysates[38]. To rule out indirect interactions, which may result from crude cell lysates, we purified components of the SWI/SNF complex from recombinant baculovirus-infected Sf9 cells. The K-Rta transactivation domain (Fig. 1e) and its mutant were expressed as GST-fusion proteins and purified from *E.coli* (Mut1, 2, and 3 in Fig. 1f, g). The GST-pull down studies using wild type (WT) K-Rta domain showed that the SWI/SNF complex containing SMARCA4, SMARCC2, and SMARCB1, directly interacted with K-Rta between residues 551 and 650. Alanine substitutions were introduced to generate three mutant K-Rta proteins referred to as Mut1, 2, and 3 (Fig. 1f), and we determined that residues 619-621 (ILQ), which are mutated in Mut2 and 3, were critical for the interaction (Fig. 1h). On the other hand, mutation of residues 612-613 (DD), which are mutated in Mut1, did not impair the interaction with the purified proteins (Fig. 1h). Interestingly, additional mutation of negatively charged amino acids (DD) to hydrophobic amino acids (AA) in Mut3 slightly restored interaction of the Mut2 protein with SMARCB1 (Fig. 1h). Mut2 has a mutation reduces hydrophobicity due to alanine substitution (i.e., ILQ > AAA). The result may indicate that the hydrophobicity of this sequence stretch plays an important role in the interaction between K-Rta and

SMARCB1. Protein structure modeling with IUPred2[43,44] showed that the interaction domain is located within an intrinsically disordered region (IDR) of K-Rta (Fig. 1e, bottom panel), and IDRs are known to be involved with transcriptional activation through flexible interactions with both nucleotides and proteins[45]. The results suggest that K-Rta utilizes the IDR to directly interact with the large coactivator complex and recruit the complex to K-Rta DNA binding sites for transactivation during lytic replication.

**K-Rta peptide inhibits leukemia and lymphoma cell growth in vitro.** To further define a protein sequence within K-Rta responsible for coactivator complex formation, homologous protein sequences were extracted from other gamma-herpesvirus homologs and a bacterial transcription factor with BLAST analysis. The consensus protein sequence was then depicted with the hypothesis that these conserved amino acid residues are important for biological functions (Fig. 2a). Based on the conserved protein sequence, we next synthesized a series of K-Rta peptides, which are tagged with the cell penetrating 10-amino acid trans-activator of transcription (TAT) sequence for intracellular delivery[46]. To protect from rapid peptide degradation by cellular proteases, we also used an unnatural D-amino acid at the N-terminus in the peptide synthesis (Fig. 2b). Cell penetration of the peptide was confirmed by using the fluorescently labeled peptide Pep 1 (Fig. 2c). The results showed that the peptide was efficiently translocated into the cell nucleus within 15 min of incubation.

We next examined the effect of the peptides on cell viability and viral replication. The hypothesis was that the introduced peptide would compete with the coactivator complex (e.g. SWI/SNF complex) as a decoy to inhibit cellular and viral gene transcription. Different concentrations of peptides were incubated with BC-1 cells, a KSHV latently infected PEL cell line, and cell viability was examined with MTT assays. The results showed that the peptides Pep 1 and Pep 3 reduced cell viability by 90% at 16 μM concentration compared to the untreated control, while a mutant peptide containing a triple alanine substitution (Mt-P) or a peptide with deletion of leucine at position 616 of K-Rta (Pep 2) only showed 20% reduction at 64 μM (Fig. 2d). Also, Pep 4 lacking the C-terminal 4 amino acids including conserved leucine at position 625 of K-Rta was ~4-fold less effective compared to Pep 1 and 3 (Fig. 2d). Notably, some substitutions of non-conserved amino acids (618S to T, 628T to E or S) did not affect the peptide's cell killing ability (Supplementary Fig. 2). In addition, including two additional D- amino acids at the C-terminus, which has been shown to increase peptide stability[47], did not enhance cell killing ability (Supplementary Fig. 2). Based

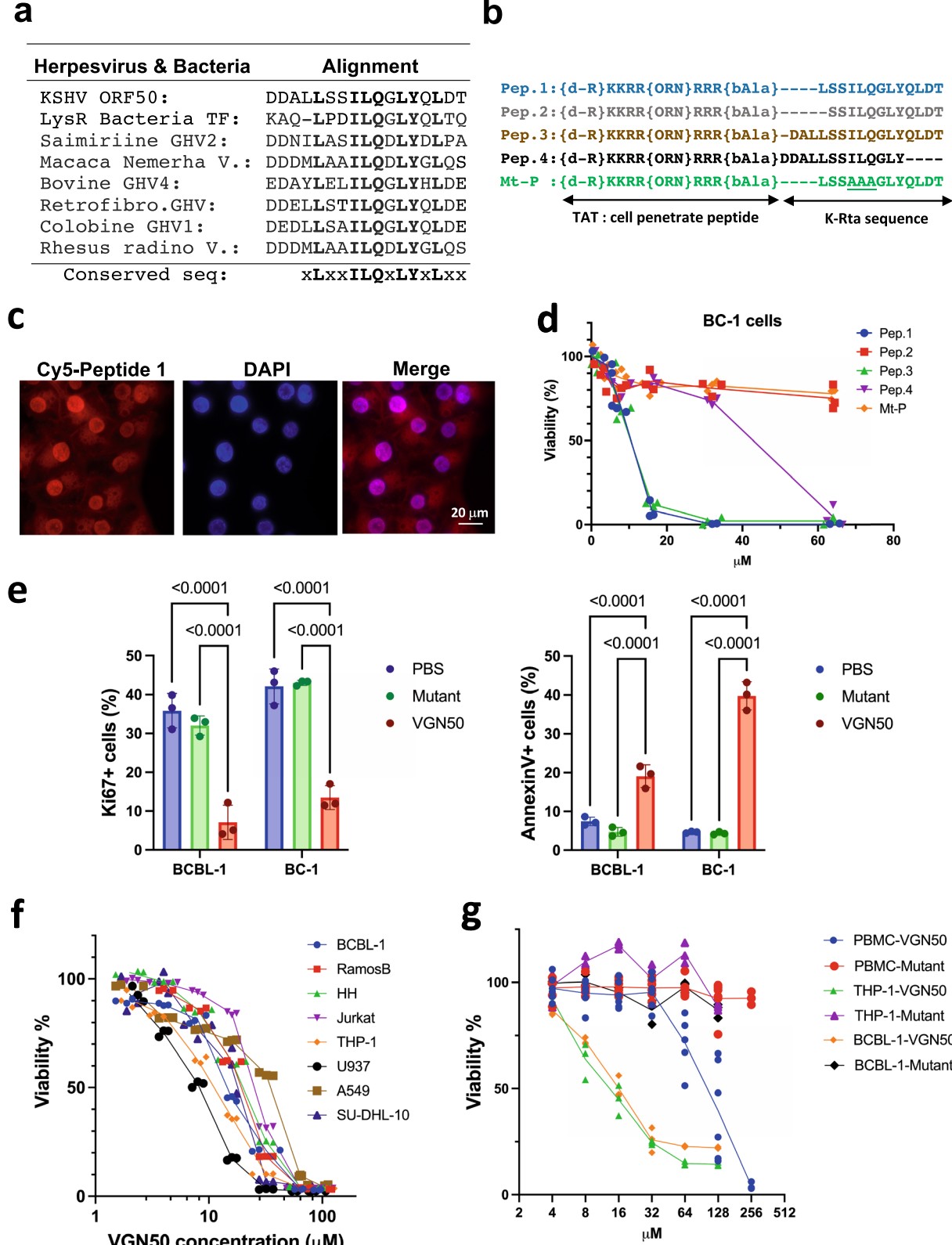

on these results, we decided to use peptide 1 (Pep 1) and its mutant (Mt-P) for the further experiments.

We next used two PEL cell lines and flow cytometric analyses to confirm that reduced cell viability was due to down-modulated cell proliferation and apoptosis induction, which was seen as early as 6 h post incubation (Fig. 2e and Supplementary Fig. 3).

Furthermore, we found that cell killing by Pep 1 was not limited to KSHV-latently infected PEL cells and was also seen in KSHV-negative myeloid and lymphoid cancer cell lines. However, the A549 adenocarcinomic human alveolar basal epithelial cell line, was 4-fold resistant to cell killing by Pep 1 (Fig. 2f). Thus, the cell killing activity is independent of KSHV latent infection.

**Fig. 2 Viral and cellular responses to K-Rta peptide; identification of VGN50. a** Protein sequence alignment. Homologous protein sequences were searched for with BLAST and extracted from other gamma-herpesviral homologs. The consensus protein sequence is depicted at the bottom of table. **b** Peptide design. Deletion peptides were designed based on the consensus sequence in **a**. The TAT protein sequence was used as a cell-penetrating peptide (CPP) and placed downstream from the K-Rta protein sequence. (d-R) D amino acid arginine, (ORN) ornithine, (bAla) beta alanine. **c** Sub cellular peptide localization. Cy5-labeled peptide 1 was used to track subcellular localization. Nuclei were visualized by DAPI staining. The scale is indicated in the panel. **d** MTT assays with deletion and mutant peptides. Peptide listed in **b** were incubated with BC-1 cells and MTT conversion was measured 48 h after incubation. The OD of mock-treated samples were set as 100%, and OD from detergent-treated cells were set as 0%. The amount of each peptide used for incubation is depicted along the x-axis. Mean percentage viability ±SD were calculated for each treatment group ($n = 3$ samples). Peptide 1 was renamed as VGN50. **e** Flow cytometry analyses of cell proliferation and apoptosis induction. Cells were incubated with VGN50 or mutant peptide (three alanine substitutions) for 24 h followed by measurements of percentage cell proliferation (Ki67 staining) and apoptosis induction (Annexin V staining). Standard deviation bars are included. ($n = 3$) Two-way Anova, followed by Sidak's multiple comparisons test. **f** The effect of VGN50 on various cancer cell types. MTT assays were performed with the indicated cell lines treated with various VGN50 concentrations. The OD of mock-treated samples were set as 100%, and OD from detergent-treated cells were set as 0%. Mean percentage viability ±SD was calculated for each treatment ($n = 3$ samples/treatment). **g** Viability assay with flow cytometry. Cell viability was measured in triplicate with live/dead staining and the cell killing effects on cancer cells was compared with normal peripheral blood mononuclear cells (PBMC) from three healthy donors. Results are presented as mean percentage viability ±SD ($n = 3$ samples/group). **f**, **g** Ordinary one-way Anova, followed by Tukey's multiple comparisons test.

Importantly, PBMCs were approximately 10-fold less sensitive to Pep 1-mediated killing compared to myeloid and lymphoid cancer cell lines as assessed by flow cytometry after live/dead staining (Fig. 2g). Based on the oncolytic activity, we renamed the K-Rta Peptide 1 sequence VGN50 (<u>V</u>irus de <u>G</u>ann wo <u>N</u>aosu OR<u>F50</u>), which means curing cancers with viral protein(s) in Japanese.

**VGN50 binds SWI/SNF proteins and mimics K-Rta function to induce KSHV lytic gene expression.** Because VGN50 is derived from the KSHV transactivator sequence, we next examined if VGN50 can inhibit KSHV lytic replication by competitively blocking the coactivator binding to K-Rta. To study this, we used the TREx-K-Rta BCBL-1 cell line, in which we can trigger the expression of exogenous Flag-tagged K-Rta in a doxycycline (Dox) inducible manner. TREx-K-Rta BCBL-1 cells were treated with 12 μM VGN50 or mutant peptide, with or without Dox induction. We used approximately one-half of the IC50 concentration of VGN50 to treat the TREx-K-Rta BCBL-1 cells in order to avoid cell toxicity effects that would non-specifically decrease viral gene expression. KSHV gene expression was profiled with a KSHV genome-wide PCR array, using ribosomal rRNA for normalization and as an internal control. Fold activation relative to the uninduced Mut-P-treated sample (not induced: Dox -) was determined and a heatmap for KSHV lytic gene expression was generated (Fig. 3a). The results showed that, in both Mut-P- and VGN50-treated cells, KSHV lytic gene expression was increased with Dox incubation. VGN50-treated cells with Dox showed the highest expression, indicating that contrary to our expectation, VGN50 enhanced the KSHV lytic program. Immunoblotting was used to confirm the induction of lytic replication by Dox based on exogenous K-Rta protein (Flag-K-Rta) and the expression of two other KSHV lytic proteins (ORF57 and K-bZIP) along with actin and KSHV LANA protein as controls for equal loading and similar viral copy numbers in samples, respectively (Fig. 3b). Notably, VGN50 alone weakly triggered PAN RNA expression, an indication of KSHV reactivation, in KSHV-latently infected BCBL-1 and BC-1 cells (Fig. 2c), which is consistent with enhanced lytic gene expression seen in K-Rta inducible cells (Fig. 2a). Taken together, these results indicate that VGN50 is a biologically active molecule that mimics K-Rta function but fails to block K-Rta function under the current conditions.

To confirm the basis of the VGN50 action, the biochemical interactions between VGN50 and SWI/SNF proteins (i.e., putative VGN50 target molecules), were examined by ELISA-based binding assays using purified five individual SWI/SNF components prepared from baculovirus-infected Sf9 cells (Fig. 3d). Increasing concentrations of biotin-conjugated VGN50 or Mut-P were incubated in an ELISA plate coated with each SWI/SNF component, and the bound peptides were detected by HRP-streptavidin (Fig. 3e). A high concentration (1 mg/mL) of bovine serum albumin was used for blocking and in every incubation step to reduce non-specific interactions. The results showed that VGN50 bound to the 5 SWI/SNF components at a concentration as low as 50 nM at different efficacies (Fig. 3f). VGN50 seemed to have slightly better binding affinity to SMARCA4, SMARCB1, and SMARCE1, whereas the Mut-P control peptide did not bind SWI/SNF components as much as VGN50 at the same peptide concentrations (Fig. 3f). These results suggest that VGN50, a prototype of the K-Rta IDR, can flexibly interact with components of the SWI/SNF complex, and that the highly conserved ILQ sequence is required for binding.

**VGN50 down-regulates MYC-mediated gene transcription.** As a next step, we sought to identify the cellular pathways that are impacted by VGN50. For this, RNA-seq analysis was conducted on total RNA isolated from cells harvested after 24 h of incubation with VGN50. Mutant peptide was used as a comparison to identify differentially regulated genes by VGN50 (Fig. 4a). Consistent with the fact that VGN50 is derived from K-Rta, and that one of the recognized K-Rta functions is to counteract host interferon responses, gene set enrichment analyses (GSEA) showed that both α- and γ- interferon pathways associated gene sets were clearly down-regulated by the peptide incubation (Fig. 4b). GSEA also showed that MYC target genes, which include enzymes associated with DNA replication, were among the significantly down-regulated gene sets (Fig. 4a, b). Furthermore, we confirmed the result by RT-qPCR that the expression of *MYC* itself was decreased in both BCBL-1 and BC-1 cells after treatment with VGN50, but not mutant peptide (Fig. 4c).

In order to reveal direct targets of VGN50, we next employed the thiol(SH)-linked alkylation for the metabolic sequencing of RNA (SLAM seq) method[48]. SLAM-seq is an orthogonal-chemistry-based RNA-Seq technique that detects 4-thiouridine (s4U) incorporation in RNA species at single-nucleotide resolution. We separated newly synthesized RNAs that are transcribed in the presence of s4U in tissue culture media from existing RNAs based on sequence reads containing T > C conversions as a result of the alkylated s4U's. We also replicated the experimental conditions used in comprehensive target profiling studies of the BET bromodomain inhibitor with SLAM-seq[9] and collected samples at the same time points (Fig. 4d), which allowed us to compare target gene profiles between VGN50 and the BET

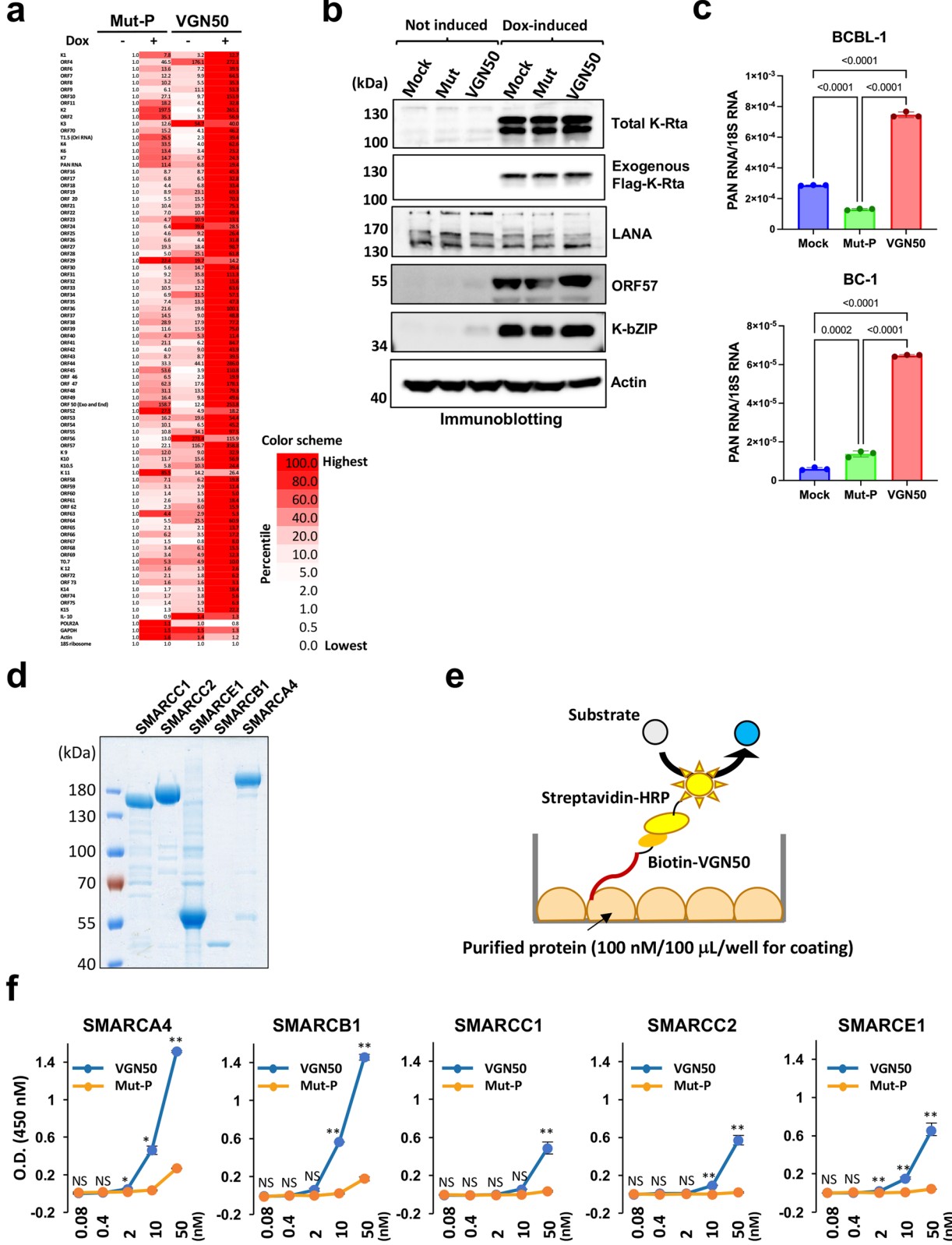

bromodomain inhibitor, JQ1. The BET bromodomain inhibitor is an epigenetic drug, which is also known to down-regulate MYC expression. Consistent with the total RNA-seq results, *MYC* exhibited the most statistically significant down-regulation among 25,000+ cellular genes in BCBL-1 with a logFC of −2.15 (FDR-adjusted *p*-value of 9.51 E-12). Similarly, BC-1 cells also displayed a log2FC −1.643 (FDR-adjusted *p*-value 5.78E-12) in MYC

expression (Supplementary Data 2 and 3). The Integrative Genomics Viewer (IGV) snapshot of the *MYC* 3'-UTR region showed that treatment with VGN50 down-regulated active *MYC* transcription (i.e., nascent transcripts) in BC-1, while pre-existing MYC mRNA species (i.e., non T->C) showed little changes (Fig. 4e). A primary effect of VGN50 is clearly gene down-regulation, as evidenced by the majority of statistically significant

**Fig. 3 Effects of VGN50 on KSHV gene transcription and VGN50 binding with SWI/SNF components. a** KSHV qRT-PCR array analysis was used to study effects of VGN50 on KSHV gene transcription in a genome-wide manner. Uninduced (−; no doxycycline added) or dox-induced (+; 1 μg/ml) cells were treated with VGN50 or the mutant peptide (Mut-P) at a concentration of 24 μM. Samples were collected at 24 h post-treatment. Expression in uninduced cells was set as 1 for each viral gene and expression shown as fold activation relative to no-Dox samples. Selected cellular genes were also included in the PCR array for comparison. 18S ribosomal RNA was used as an internal control. **b** KSHV protein expression. Immunoblotting was performed to confirm both endogenous and exogenous K-Rta expression before (not induced) and after induction by Dox treatment (Dox-induced). Latent KSHV protein LANA and actin were used as controls for KSHV copy number (LANA) and loading (actin), respectively. Protein samples were collected 48-h post Dox incubation. **c** KSHV reactivation by VGN50 incubation. BCBL-1 and BC-1 cell lines were treated with VGN50 (16 μM concentration) and total RNA harvested at 24 h post treatment. KSHV PAN RNA expression was examined by qRT-PCR and relative expression over 18S rRNA is shown. Data are presented as mean ± SD (n = 3). **d** Purification of recombinant SWI/SNF components. SDS-PAGE analysis of five SWI/SNF components individually prepared from baculovirus-infected Sf9 cells. **e** A schematic illustration of ELISA assay to evaluate VGN50 and SWI/SNF interaction. **f** Analysis of VGN50 binding to SWI/SNF components by ELISA. Increasing concentrations of biotin-conjugated VGN50 or Mut-P were incubated in duplicate in an ELISA plate coated with each SWI/SNF component. Peptide binding measured as OD values at 450 nm are shown. Mean OD values were compared between the VGN50 and Mut-P in each concentration using unpaired *t*-test. \*\**p* < 0.01, \**p* < 0.05, NS no significance. Data are presented as mean ± SD with duplicated samples and repeated three times with different purified protein preps.

gene expression changes (*p* < 0.05, red dots) with negative fold changes (Fig. 4d). The results are in good agreement with a previous study, which demonstrates that the KSHV transactivator takes the endogenous host cellular transcription machinery away for KSHV replication[36]. GSEA of the VGN50 direct targets were also performed, and the analyses found that MYC target genes, including MYC itself, were clearly enriched in BC-1 cells with a normalized enrichment score (NES) of 2.90 (Fig. 4f). Several cellular genes were commonly up-regulated between BC-1 and BCBL-1, and those genes were associated with inflammatory responses and immediate-early genes (e.g. JUND, FOS, ATF, Supplementary Data 4). As expected, negligible gene expression was altered by incubation with the mutant peptide in both BCBL-1 and BC-1 lines, attesting to the peptide-sequence specific regulation (Fig. 4d).

Preferential targeting of *MYC* by VGN50 also prompted us to compare target genes with the BET bromodomain inhibitor. We previously demonstrated that the BET bromodomain inhibitor JQ1 could induce KSHV reactivation similarly to that of VGN50 (Fig. 3a, c), and also down-regulated *MYC* and MYC-target genes in KSHV-positive PEL cell lines[49]. Publicly available SLAM-seq data sets for K562 (chronic myelogenous leukemia cells) were used to compare direct target gene sets with VGN50-treated PEL cell lines. The top 100 down-regulated genes were extracted and examined for similarity among down-regulated genes. The results showed that very few cellular genes were commonly down-regulated between the BET bromodomain inhibitor JQ1 and VGN50, and target gene sets were mostly dependent on cell lines. However, one of the three common target genes was *MYC* (Fig. 4g).

Finally, using CSCAN[50], we examined enrichment for transcription factors that are bound to promoters down-regulated by VGN50. CSCAN is a ChIP-seq database to identify common regulators that recognize selected gene promoters[50]. For this CSCAN analysis, we used the database for GM12878, a lymphoblastoid cell line, which we considered to be the closest to PEL cells among the available datasets. The results again showed that MYC and also IRF4, an upstream regulator of MYC in PEL cells[51,52], are enriched in VGN50 targeted promoters. The results suggested that VGN50, a peptidyl mimic of K-Rta, may appropriate the specific coactivator complex, which is also utilized by MYC and IRF4 in PEL cells. Targeted transcription factors (TFs) were largely overlapped between BC-1 and BCBL-1 (Fig. 4h), even though only 16 genes were shared amongst the top 100 down-regulated genes in common between BC-1 and BCBL-1 (Fig. 4g). In addition, TFs that are known to localize at super-enhancers such as MYC, IRF4, MEF2A, and STAT5A are commonly targeted by VGN50 in BCBL-1 and BC-1 cells (Fig. 4h,

the complete list is presented in Supplementary Fig. 4). Notably, two of three commonly down-regulated targets, *MYC* and *CCND2* (Fig. 4g), were recently found as super-enhancer-regulated genes in PEL cell lines[53].

**VGN50 inhibits coactivator complex recruitment to MYC promoter and putative enhancer regions**. We next examined the molecular mechanism of *MYC* down-regulation by VGN50. We first utilized Cleavage under targets & release using nuclease (CUT&RUN)-sequencing to examine histone modifications that are known to be associated with enhancers in order to map putative MYC enhancer regions in PEL cells (Fig. 5a). Genome-wide CUT&RUN studies identified putative enhancer and promoter regulatory regions that are marked by H3K27Ac, H3K4me1, POLR2A (RNAPII), BRD4, and SMARCC1, a component of the SWI/SNF complex (Fig. 1b and 5a). The identified putative *MYC* enhancers were also in good agreement with previous ChIP Hi-C[53] and ChIP-seq data sets[54]. Specific primer pairs were then designed to amplify the respective enhancer regions and to examine regulation of coactivator complex recruitment by VGN50. The results demonstrated that the occupancies of RNAPII, BRD4, SWI/SNF complex, along with H3K27Ac modification were decreased in the presence of VGN50 compared with Mut-peptide (Fig. 5b, c).

Since VGN50 directly binds SWI/SNF components (Fig. 3f), we hypothesize that VGN50 traps SWI/SNF components and blocks functional enhancer complex assembly. To test the hypothesis, in vitro SWI/SNF complex formation was further examined by sucrose gradient with or without VGN50. The results showed that the VGN50 peptide incubation induced the assembly of larger SWI/SNF complexes and moved the complex to heavier bottom fractions in the gradient (Fig. 5d). Immunofluorescence studies with KSHV reactivating BCBL-1 cells also showed that SMARCC1 and SMARCA4 signal intensity were increased at the KSHV transcription/replication compartment and colocalized with K-Rta protein in reactivating cells (Fig. 5e). Based on these results, we suggest a model, in which VGN50 acts as a decoy to trap the coactivator complex and block their binding to cellular enhancers, thereby preventing RNAPII to re-bind highly-inducible genomic regions (Fig. 5f). In the case of PEL cell lines, for which growth is highly dependent on the MYC and IRF4 pathways, the major targets for VGN50 were MYC and IRF4 (Fig. 4f–h).

**VGN50 inhibits tumor growth in a PEL xenograft model**. Based on strong and selective MYC down-regulation and cell killing activity of cancer cells in vitro, we next studied the

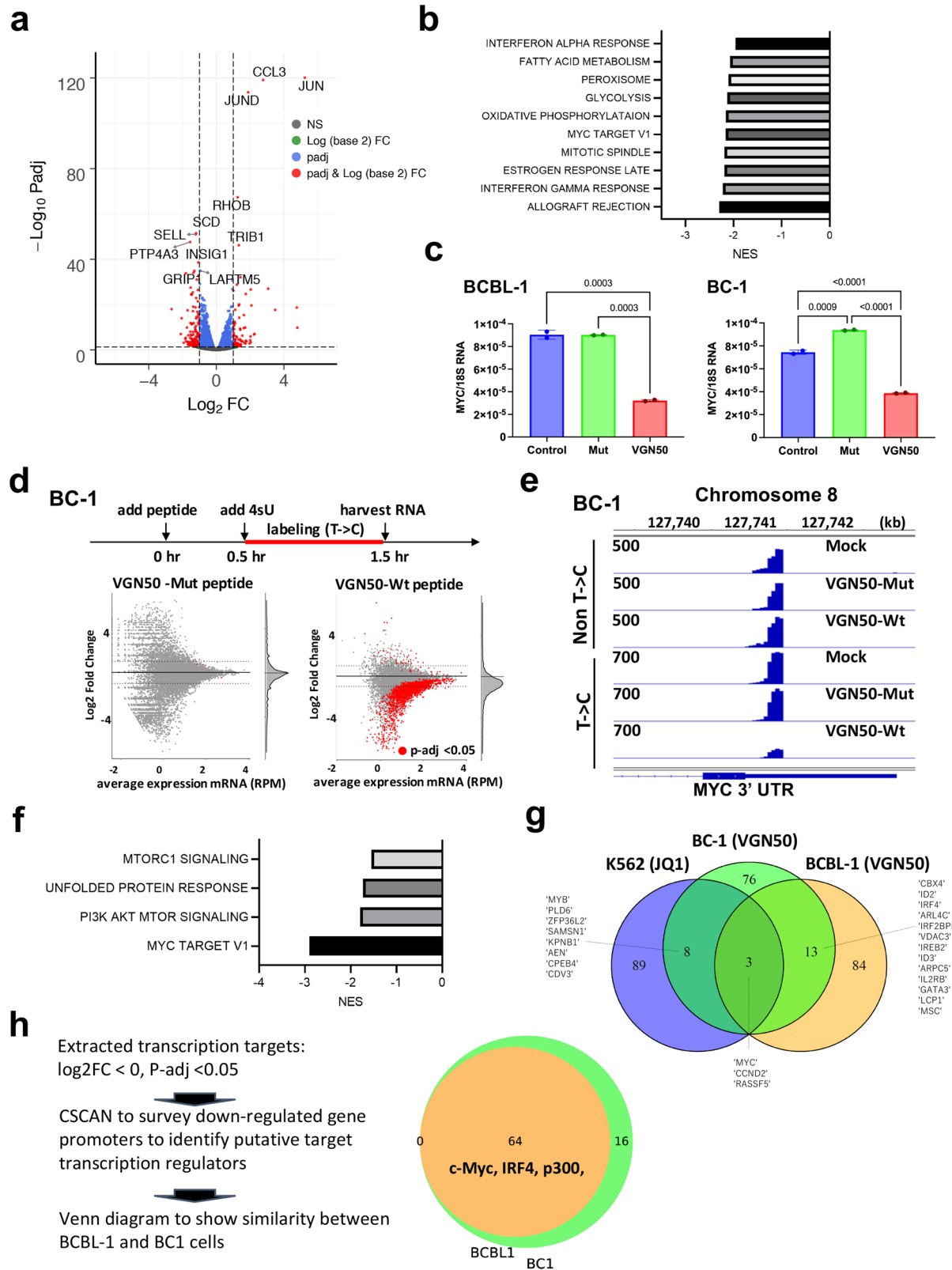

therapeutic potential of VGN50 by examining the efficacy of cancer cell killing over toxicities to mice in a PEL xenograft model with BCBL-1 cells. Pilot studies determined that the peptide was well-tolerated in immunocompetent mice when administered as 10 mg/kg peptide via intraperitoneal (i.p.) injection with a 5-day-on 2-day-off schedule for 2 weeks, as shown by no notable changes in body weight, complete blood counts, and serum

biochemical properties (Supplementary Fig. 5). Based on the drug schedule, we examined the effects of the peptide on xenograft PEL cell growth. In this established model, BCBL-1 cells grow in the peritoneal cavity resulting in the accumulation of ascites, which can be measured as a body weight gain that corresponds to the volume of ascites[55]. We inoculated male NRG mice with $2 \times 10^7$ BCBL-1 cells via i.p. injection followed by injections of VGN50

**Fig. 4 VGN50 target identification with RNA-sequencing. a** Total RNA-sequencing. BCBL-1 cells were treated with either VGN50 or mutant peptide for 24 h (24 μM, n = 3). Untreated cells were used as a negative control. RNA-sequencing datasets from VGN50-peptide-treated samples were compared with that from mutant-peptide-treated samples, and differentially regulated genes were depicted as a volcano plot (left panel). **b** Gene Set Enrichment Analyses (GSEA). GSEA was performed on differentially regulated genes and enriched, down-regulated cellular pathways are shown in the Table. **c** MYC down-regulation by VGN50. Peptide-treated samples were used for qRT-PCR analysis in duplicate and confirmed MYC down-regulation. (VGN50, 24 μM, 24 h post treatment) **d** Direct target identification with SLAM-seq. The experimental scheme is shown in the upper section of the panel. VGN50 or mutant peptide (24 μM final concentration) were added to BC-1 or BCBL-1 cells 30 mins prior to incubation with 4sU, and RNA was labeled for 1 h in the presence of each peptide. The C/T converted RNA-species were compared with mock-treated samples and depicted as scatter plots. Each treatment was performed in duplicate and nascent transcripts with p-values <0.05 are indicated as red dots. **e** Integrative Genomics Viewer (IGV). Both non-T/C converted and converted sequence reads were visualized with the IGV, and a snapshot of the MYC 3'-UTR region is presented. SLAM-seq libraries were made with dT-primers demonstrating sharp peaks at the 3'-UTR. **f** GSEA analyses for VGN50 direct targets. GSEA was performed with the group of genes that were down-regulated by VGN50 (LogFC < 0, p < 0.05). **g** Comparisons of JQ1 and VGN50 direct target genes. The top 100 down-regulated target genes for each treatment were extracted and used to generate a Venn diagram in order to examine similarity. Commonly down-regulated gene names are depicted. **h** Co-occupancies of regulatory proteins in down-modulated genes. The strategy of gene selection for CSAN is described on the left-hand side of the panel. CSCAN was applied to extract potential binding proteins on down-regulated gene promoters (−450 to +50 bp). The number of common transcriptional regulators are shown in the Venn diagram and selected transcription and co-regulatory protein names are also included. The complete list of gene names is presented in Supplementary Fig. 3.

(10 mg/kg), mutant control peptide or PBS, daily for 10 days starting from 2 days after (day 0) tumor inoculation. All mice in the control PBS-treated groups reached the humane endpoint by 10 days (Fig. 6a). However, in mice that received the VGN50 treatment, tumor growth was inhibited as evidenced by the absence of weight gain due to accumulation of ascites fluid containing tumors during the course of therapy on days 0–10 compared with the PBS or mutant control peptide-treated groups (p < 0.05; Fig. 6a). At the termination point, tumor growth was also validated by flow cytometry analysis of PEL cells in ascites fluid (Fig. 6b, c). Notably, cell size (FCS-A) and scatter (SSC-A) parameter analysis revealed that BCBL-1 cells from ascites of mice treated with VGN50 showed reduced cell size and scatter geometric mean fluorescent intensity (gMFI) compared to the control groups (Fig. 6c and Supplementary Fig. 6). The results suggest that the peptide caused cell atrophy. A similar phenotype is reported in *MYC* knock-out cardiac myocytes and keratinocytes[56,57].

Since PELs are frequently involved in effusions in multiple body cavities in human patients and the amounts of inflammatory cytokines in the effusion are strong prognostic factors[58], we also extracted ascites fluid and profiled it to see if the PEL cell-derived human cytokine profile was also altered by continued VGN50 treatment. Principal component analysis of the expression of 15 inflammatory proteins that are detected in ascites fluids among 92 total proteins (Supplementary Data 5) screened showed clear differences between the treatment groups and that clearly separated VGN50-treated cells from those of control groups (i.e. PBS- or mutant peptide-treated) (Fig. 6d). Among 15 cytokines differentially regulated, a large fraction of cytokine secretion, except for a few including IL-8, was down-regulated in VGN50-treated tumors (Fig. 6e and Supplementary Fig. 7). The cytokine profile was also in good agreement with the RNA-seq results from PEL cells in xenograft mice, which demonstrated that genes involved in TNFα signaling with the NF-κB pathway were the most down-regulated gene sets in PEL xenograft cells treated with the peptide (NES 1.74, p < 0.01) (Supplementary Fig. 6). Although it was not as clear as in vitro tissue culture studies, MYC target genes were again among the enriched down-regulated gene sets (Supplementary Fig. 8). Taken together, these results suggest that the VGN50 should have therapeutic value to control PEL pathogenicity.

## Discussion

We are very excited to report the identification of VGN50 derived from a viral protein sequence, which has potential as a

therapeutic module to regulate MYC expression. The peptide sequence is derived from the intrinsically disordered domain of the extremely potent viral transactivator protein, K-Rta[29,59,60]. Even though the transactivation domain is predicted to be highly disordered, the 13-amino acid sequence stretch is well-conserved among other gamma-herpesviruses and a bacterial transcription factor, suggesting that the sequence is intended and well-designed to be non-structured evolutionarily for transactivation function (Fig. 2a, b).

Previous studies demonstrated that K-Rta inhibits IFN-mediated cellular responses by degrading IRF7[61,62]. Here we show that introduction of a small fragment of the K-Rta transactivation domain attenuated IFN target gene transcription (Fig. 4b), suggesting that K-Rta regulates IFN activation both transcriptionally and post-translationally. Enrichment of IFNγ target genes also suggests an interesting possibility that K-Rta may take advantage of host IFN responses for its own gene expression by hijacking actively-assembling transcription coactivators that are specialized for transient and robust IFN target gene expression. We wonder if primarily targeting IFN pathways by K-Rta is the reason for VGN50 being more efficacious against myeloid and lymphoid cells. In these myeloid and lymphoid cell types, regulation of cell growth is more directly associated with host immune responses for replication.

Recent studies showed that another KSHV protein, vIRF3 activates IRF4 super-enhancer and activates downstream targets of *MYC* expression during latency[52], while the lytic KSHV protein, viral IRF4, inhibits cellular IRF4 binding to the *MYC* enhancers[54,63], therefore, inhibiting *MYC* expression during lytic replication. It is known that KSHV reactivation has a Yin and Yang relationship with cellular *MYC*; *MYC* expression inhibits KSHV reactivation while knock-down of *MYC* or *IRF4* reactivates KSHV in latently infected PEL cells (Fig. 3c and refs. [64,65]). A similar relationship was also reported recently for Epstein-Barr virus infected B-cells[66]. The authors elegantly performed genome-wide CRISPR-Cas9 screening and identified that MYC and MYC-activating proteins are necessary for maintaining EBV latency in B-cells[66]. These studies indicated that latent gamma-herpesviruses sense the amount of MYC in the host cell nucleus for reactivation. Our study hypothesizes that it is, in part, due to competition for the NCoA2-SWI/SNF chromatin remodeling complex. We hypothesize that overexpression of MYC associates with a limited supply of co-activator enzymes that are needed for K-Rta to activate viral promoters, while a decrease in the total amount of MYC may increase available resources and chances for K-Rta to assemble coactivator complex on KSHV gene promoters. Interestingly, incubation of cells with VGN50 at one-half the IC50

 

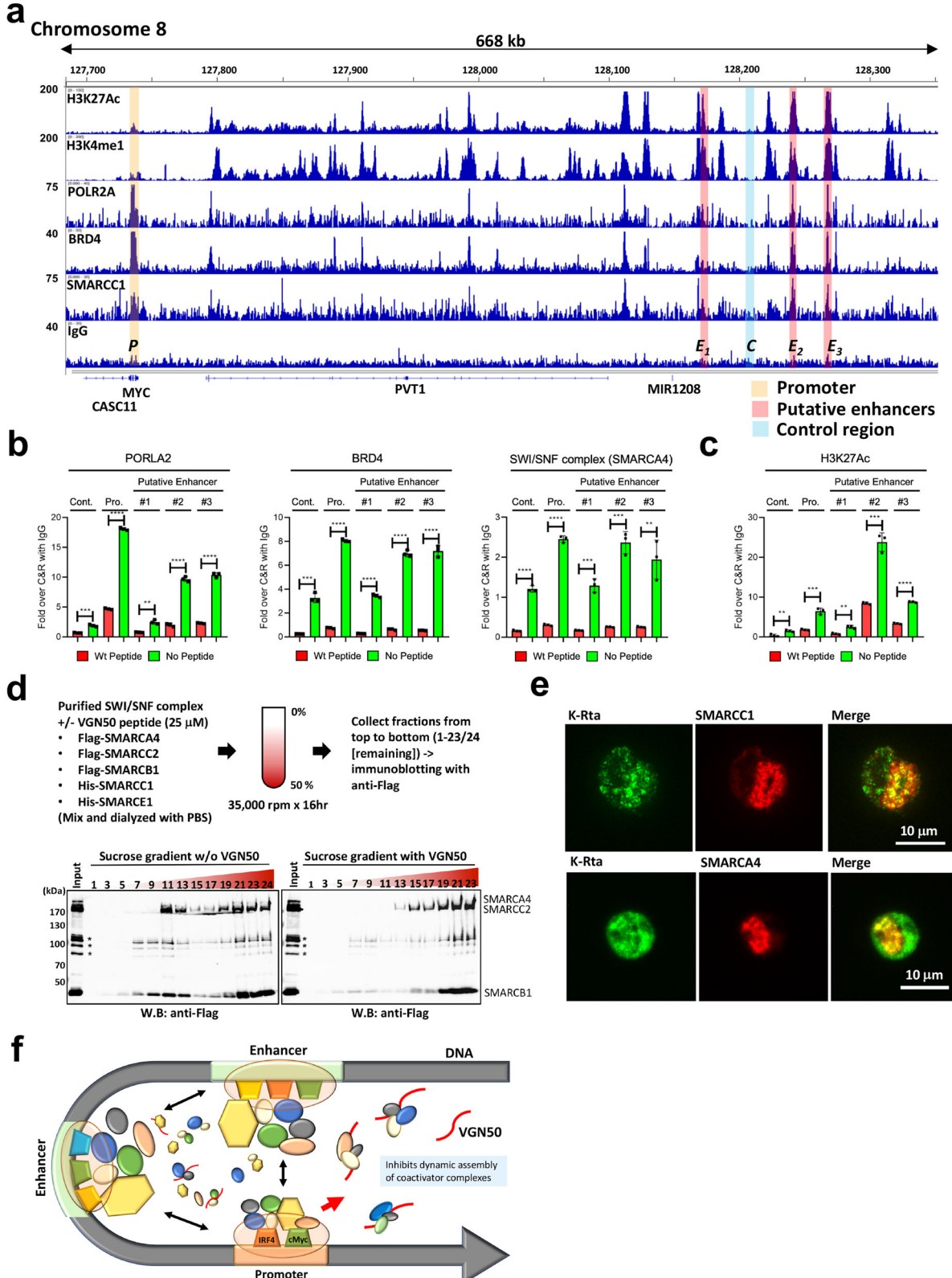

dose did not inhibit KSHV lytic gene expression, but instead further increased the overall KSHV transcripts. We speculate that such concentration of VGN50 may still inhibit coactivator binding to cellular transcription factors (i.e. MYC, IRF4), but not KSHV K-Rta. K-Rta is known to form oligomer[67] and may require higher concentration of VGN50 in order to successfully compete with K-Rta protein.

MYC expression is tightly regulated in normal cells, and tumor cells are often addicted to MYC expression for their growth due to great demand for continued generation of metabolites, thus being more sensitive to MYC inhibition (Fig. 2e, f). Transgenic expression of Omomyc, a dominant-interfering MYC mutant, indeed showed remarkably low toxicity to mice despite well-documented effects of systemic MYC inhibition on normal

**Fig. 5 A molecular mechanism of VGN50-mediated MYC-down-regulation. a** Cleavage under targets & release using nuclease (CUT&RUN). CUT&RUN was performed with BCBL-1 and BC-1 with the indicated antibodies. Enrichment of active histone modifications, and occupancies of RNAPII and SWI/SNF components at the MYC genomic region are shown as peaks. One of the biological duplicate samples for the BCBL-1 cell line is presented. BC-1 showed very similar peak distributions for these molecules. The MYC promoter, enhancer, and negative control regions used for qPCR analyses are indicated with shading. The positions of the MYC coding region, CASC11 coding region, and long non-coding RNA, PVT1, MIR1208 are shown in the panel. **b** VGN50 inhibits coactivator complex recruitment and **c** reduces H3K27Ac modification. qPCR is used to examine enrichment of DNA fragments in the presence of VGN50 or mutant peptide (24 μM). Peptide was applied 1 h prior to processing samples for CUT&RUN to replicate SLAM-seq experimental conditions. Non-specific IgG control was also included and enrichment with IgG control at each genomic region was set as 1 for b, and spiked-in luciferase plasmid DNA was used for additional internal control for c. **d** Formation of a larger protein complex with VGN50. Sucrose gradient sedimentation was used to monitor changes in protein complex formation in the presence of VGN50. The experiment flow chart is shown at the top of this panel. Fractions were collected from top (#1) to bottom (#23/24). Sample #24 without peptide is a smaller fraction of residual samples. Immunoblotting was performed on the odd number samples and the remaining fraction (#24 for no peptide sample) and probed with Flag-tagged SWI/SNF components (SMARCA4, SMARCC2, and SMARCB1). The position of each protein component based on molecular size is indicated. **e** KSHV K-Rta colocalizes with SWI/SNF components in reactivating BCBL-1 cells. TREx-BCBL-1 cells were triggered for reactivation by treatment with Dox and TPA for 28 h and stained with the indicated antibodies. Images were acquired with Keyence fluorescence microscopy. **f** A model for VGN50 molecular action. A model depicting the putative molecular action of VGN50 is presented. Coactivator complexes are dynamically assembled on enhancer and promoter regions via transcription factor binding. VGN50, an IDR fragment derived from an exceptionally potent viral transactivator, flexibly interacts with components of coactivators and traps them, thus reduces the available resources for cellular enhancers to activate MYC promoters.

regenerating tissues[68]. Consistent with this, we observed that the EC50 of VGN50 for PBMCs from healthy donors was approximately 10-fold higher than that for cancer cell lines (Fig. 2g), and VGN50 administration to immune competent mice at 10 mg/kg with a 5-day-on 2-day-off schedule for 2 weeks was well-tolerated without substantial loss of weight (Supplementary Fig. 5).

Human SWI/SNF complexes are remarkably large, including the products of the 29 genes encoding mSWI/SNF subunits that assemble into three distinct mSWI/SNF complexes, termed canonical BRG1/BRM associated factor (cBAF), polybromo-associated BAF (PBAF), and noncanonical BAF (ncBAF), each of which comprises common as well as complex-specific subunits[69]. Recent studies have revealed a high prevalence of mutations in genes encoding subunits of the SWI/SNF complexes in cancers and neurological diseases[69,70], indicating the importance of this complex as a therapeutic target. In this context, our current study showed that the oncogenic KSHV utilizes the SWI/SNF complex for the viral genome replication through the interaction with the viral transactivator protein K-Rta. Moreover, we showed that VGN50, a K-Rta-derived peptide, can be used to prevent the SWI/SNF complex from forming the MYC transactivation machinery in cancer cells. The limitation of the current study is that we could not pinpoint the specific molecular target and structural mechanism through which VGN50 interacts with this large SWI/SNF complex. By ELISA-based assays, we attempted to show that VGN50, but not a control mutant peptide which lacks the highly conserved ILQ sequence motif, biochemically binds to several isolated shared subunits of SWI/SNF complexes. However, further studies are required using other techniques and approaches to fully understand the structural mechanism in which this small viral-derived peptide can flexibly interact with several SWI/SNF components and capture subunits of SWI/SNF, thus preventing it from assembling a transactivation complex on the MYC promoter. The latter is especially important to better understanding SWI/SNF biology and to identify new approaches for targeting SWI/SNF complexes in cancer therapy.

In summary, our study demonstrated that a viral protein is a unique starting material as the basis for designing therapeutics directed at attenuating cellular protein function(s). We hypothesize that viral proteins continuously evolved structure possessing enhanced efficiencies to hijack cellular protein function, and resulting in the current conserved amino acid sequence. Future studies will likely increase utilities of the VGN50 peptide by targeting specific cell types as an antibody dependent conjugate or a ligand fusion protein, which may enhance the efficacy of existing antibody drugs. We hope that VGN50 can contribute to efforts to attenuate the progression of diseases, in which MYC activation plays a pathogenic role.

## Materials and methods

**Cell culture**. The BCBL-1 cell line was obtained from Dr. Ganem (University of California San Francisco). The Flag-HA-tagged-K-Rta-inducible, TREx-K-Rta BCBL-1 cell line was generated according to methods previously described[29]. The BC-1, BC-3, Ramos, HH, Jurkat, THP-1, U937, A549, and SU-DHL-10 cell lines were obtained from ATCC, expanded, and aliquots stored in order to maintain an inventory of early passage stocks. These cell lines were cultured in RPMI 1640 medium supplemented with 15% FBS, antibiotics, and L-glutamine, or DMEM medium supplemented with 10% FBS, antibiotics, and L-glutamine for A549 cell lines.

**In vitro cell treatment with VGN50**. Leukoreduction system chambers (LRS) from healthy donors were purchased from Vitalant. Peripheral blood mononuclear cells (PBMCs) were prepared by a standard Ficoll gradient method. PBMCs along with THP-1 and BCBL-1 cell lines ($2 \times 10^6$ cells/ml, 0.2 ml 10% FBS/RPMI/DMEM per well of a 96-well plate) were treated with various doses (0–256 μM) of either VGN50 peptide or control mutant peptide for 24 h. Cell death, apoptosis, and proliferation were analyzed by flow cytometry (BD Acuri) after intracellular staining with anti-humanKi67-Alexa488 (Biolegend, 1:100) along with the isotype control antibody, annexin-V-FITC with 7-AAD staining (Biolegend) according to the manufacture's protocol, Live/dead Fixable Red Dead Cell staining kit (Invitrogen) according to the manufacture's protocol, or MTT assay. Peptides were commercially synthesized (GenScript) with >90% purity with TFA removal. Peptides were dissolved in PBS and used at various concentration indicated in figure legends.

**Cytokine measurement**. PEL-derived cytokine profiles in ascites were measured with Olink analysis service with the Olink INFLAMMATION panel using proximity extension technology, a high-throughput multiplex proteomic immunoassay[71]. In short, the panel includes 92 immune-related proteins, primarily cytokines and chemokines. The assay utilizes epitope-specific binding and hybridization of a set of paired oligonucleotide antibody probes, which is subsequently amplified using quantitative PCR, resulting in log base 2-normalized protein expression (NPX) values. The data was processed and analyzed with Olink Insight Stat Analysis software.

**Rapid immunoprecipitation mass spectrometry of endogenous protein**. TREx-Flag-tagx3-HA-tagx3 (F3H3)-K-Rta BCBL-1 cells were left untreated or KSHV reactivation was induced by adding doxycycline (1 μg/mL) and TPA (20 ng/mL) in the culture media for 28 h. Cells were fixed according to the company's recommendation (Active Motif) by adding one tenth volume of formaldehyde solution into the culture medium [11% methanol-free formaldehyde (ThermoFisher), 100 mM NaCl, 1 mM EDTA, 50 mM Hepes (pH 7.9)] and incubation for 8 min. The reaction was stopped by adding glycine to the culture medium to a 100 mM final concentration for 5 min. Cells were washed three times with cold 0.5% NP-40/PBS and cell pellets were snap-frozen on dry ice. Samples were then shipped to Active Motif (Carlsbad, CA) for their Interactome Profiling Service. Antibodies used for RIME assays were anti-Flag M2 for K-Rta (Sigma), anti-RNAPII (4H8, Millipore Sigma), and control IgG provided by the company. Each

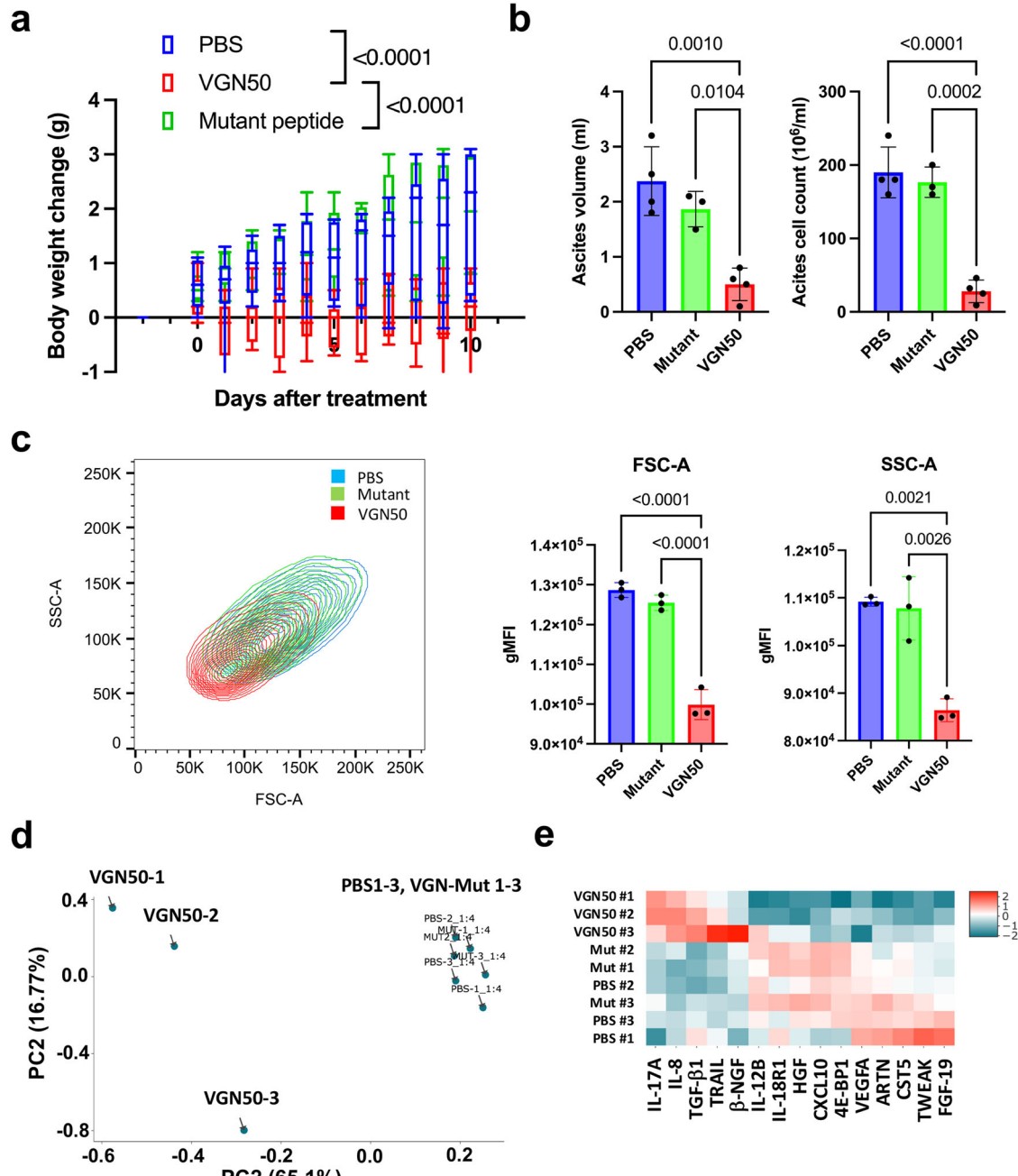

**Fig. 6 VGN50 inhibits BCBL-1 cell growth in xenograft mice. a** PEL-xenograft. BCBL-1 cells ($2 \times 10^7$ cells) were injected i.p. Peptide (10 mg/kg) was administrated i.p. daily and tumor growth was monitored by measurement of body weight ($n = 6$–9 mice per group). **b** Ascites volume and PEL cell counts. In a separate experiment, ascites volume and PEL cell counts in ascites fluid were measured at day 10. ($n = 3$–4 mice per group) **c** Induction of atrophy. BCBL-1 cells isolated from ascites fluid were measured for FSC-A and SSC-A by flow cytometry. VGN50-treated PEL cells exhibited decreased cellular volume. ($n = 3$) **d** Expression of inflammatory cytokines. Principal component analysis based on expression levels of inflammatory cytokine contents in ascites fluid. **e** Cytokine heat map. Heatmap generated based on abundance of the indicated cytokines in ascites fluids. Ascites fluid was taken at day 10. **a**–**c** Data were analyzed using ordinary one-way Anova followed by Tukey's multiple comparisons test.

immunoprecipitation was performed in duplicate. The following criteria were used to designate specific interactions: (i) more than one peptide count in each replicate sample, (ii) more than 5 total peptide counts by combining two replicates, (iii) more than 5-fold enrichment over background noise (IgG), and (iv) In addition, scaffold software was used to determine the statistical significance of the interactions and a $p$-value <0.05 was considered significant. Filtered data sets from RIME with RNAPII and K-Rta were then combined and increased or decreased interactions with commonly interacting proteins between RNAPII and K-Rta by KSHV reactivation was determined using 1.5-fold as a cutoff (Supplementary Data 1). A total 87 proteins were identified as K-Rta interacting proteins of which 85 were also identified as RNAPII-interacting proteins. Non-processed raw data sets are provided upon request.

**PEL xenografts**. All animal studies were conducted according to a UC Davis Institutional Animal Care and Use Committee (IACUC)-approved protocol. NRG (NOD.Cg-*Rag1*^*tm1Mom*^ *Il2rg*^*tm1Wjl*^/SzJ) mouse breeding pairs were purchased from the Jackson Laboratory and the colony was maintained in-house. Eight-week-old male NRG mice were injected intraperitoneally (i.p.) with $2 \times 10^7$ BCBL-1 cells in 200 µL PBS. On day 2, mice were randomly assigned to PBS control, VGN50 (10 mg/kg), or mutant control peptide (10 mg/kg) groups. Treatments were administered by daily i.p. injection for 10 days. Mice were monitored daily for PEL burden based on body weight increment. Some mice were killed for ascites cytokine measurement and PEL cell analysis by flow cytometry using a BD LSRFortessa™ instrument (BD Biosciences) and the data analyzed with FlowJo v10.8.0 (Tree Star) software.

**Cleavage Under Targets and Release Using Nuclease**. CUT&RUN[72] was performed essentially by following the online protocol developed by Dr. Henikoff's lab with a few modifications to fit our needs. Cells were washed with PBS and wash buffer [20 mM HEPES-KOH pH 7.5, 150 mM NaCl, 0.5 mM Spermidine (Sigma, S2626), and proteinase inhibitor (Roche)]. After removing the wash buffer, cells were captured on magnetic concanavalin A (ConA) beads (Polysciences, PA, USA) in the presence of $CaCl_2$. Beads/cells complexes were washed three times with digitonin wash buffer (0.02% digitonin, 20 mM HEPES-KOH pH 7.5, 150 mM NaCl, 0.5 mM Spermidine and 1x proteinase inhibitor), aliquoted, and incubated with specific antibodies in 250 μL volume. The antibodies and concentrations used in this study were: mouse anti-RNA polymerase II (Millipore, clone CTD4H8; 1:100), rabbit monoclonal anti-H3K4me1 (Cell Signaling, D1A9; 1:50), rabbit monoclonal anti-H3K27Ac (Cell Signaling, D5E4; 1:50), rabbit monoclonal anti-BRD4 (Cell Signaling, E2A7X; 1:50), rabbit monoclonal anti-SMARCC1 (Cell Signaling, D7F8S), and rabbit polyclonal anti-SMARCA4 (EpiCypher, 13-2002). After incubation, unbound antibody was removed by washing with digitonin wash buffer three times. Beads were then incubated with recombinant Protein A/ G–Micrococcal Nuclease (pAG-MNase), which was purified from *E.coli* in 250 μl digitonin wash buffer at 1.0 μg/mL final concentration for 1 h at 4 °C with rotation. Unbound pAG-MNase was removed by washing with digitonin wash buffer three times. Pre-chilled digitonin wash buffer containing 2 mM $CaCl_2$ (200 μL) was added to the beads and incubated on ice for 30 min. The pAG-MNase digestion was halted by the addition of 200 μl 2× STOP solution (340 mM NaCl, 20 mM EDTA, 4 mM EGTA, 50 μg/ml RNase A, 50 μg/ml glycogen). The beads were incubated with shaking at 37 °C for 10 min in a tube shaker at 500 rpm to release digested DNA fragments from the insoluble nuclear chromatin. The supernatant was collected after centrifugation (16,000×g for 5 min at 4 °C) and placed on a magnetic stand. DNA was extracted using the NucleoSpin Gel & PCR kit (Takara Bio, Kusatsu, Shiga, Japan). Sequencing libraries were then prepared from 3 ng DNA with the Kapa HyperPrep Kit (Roche) according to the manufacturer's standard protocol. Libraries were multiplex sequenced (2 × 150 bp, paired-end) on an Illumina HiSeq 4000 sequencing system to yield ~15 million mapped reads per sample. With separate replicated experiments, qPCR was used to examine enrichment at selected genomic regions. Primer sequences are provided in Supplementary Data 6.

CUT&RUN sequence reads were aligned to the human GRCh38/hg38 reference genome assembly and reference KSHV genome sequence (Human herpesvirus 8 strain: GQ994935.1) with Bowtie2[73]. Model-based Analysis of ChIP-seq (MACS2) was used for peak detection[74] utilizing the parameters described in the developer's manual. Peaks and read alignments were visualized using the Integrated Genome Browser (IGB)[75].

**RNA-sequencing**. Indexed, stranded mRNA-seq libraries were prepared from total RNA (100 ng) using the KAPA Stranded mRNA-Seq kit (Roche) according to the manufacturer's standard protocol. Libraries were pooled and multiplex sequenced on an Illumina NovaSeq 6000 System (150-bp, paired-end, $>30 \times 10^6$ reads per sample).

RNA-Seq data was analyzed using a Salmon-tximport-DESeq2 pipeline. Raw sequence reads (FASTQ format) were mapped to the reference human genome assembly (GRCh38/hg38, GENCODE release 36) and quantified with Salmon[76]. Gene-level counts were imported with *tximport*[77] and differential expression analysis including Volcano plot were performed with DESeq2[78].

**SLAM-seq**. SLAM-seq[48] was performed using the SLAMseq Kinetics Kit (Lexogen GmbH, Vienna, Austria) according to the manufacturer's standard protocol. Briefly, biological replicate cultures of BCBL-1 cells or BC-1 cells were incubated with VGN50 peptide (24 μM) for 30 min. Subsequently, 4-Thiouridine (s4U; 300 μM) was added to the media and the cells incubated for 1 h in order to label newly synthesized RNA. Total RNA was isolated and then the 4-thiol groups in the s4Uracil-labeled transcripts were alkylated with iodoacetamide (IAA). QuantSeq 3' mRNA-Seq (FWD) (Lexogen, Inc.) Illumina-compatible, indexed sequencing libraries were prepared from alkylated RNA samples (100 ng) according to the manufacturer's protocol for oligo(dT)-primed first strand cDNA synthesis, random-primed second strand synthesis, and library amplification. Libraries were multiplex sequenced (1 × 100 bp, single read) on an Illumina HiSeq 4000 sequencing system.

SLAM-Seq datasets were analyzed using the T > C conversion-aware SLAM-DUNK (Digital Unmasking of Nucleotide conversion-containing k-mers) pipeline utilizing the default parameters[48,79]. Briefly, nucleotide conversion-aware read mapping of adapter- and poly(A)-trimmed sequences to the human GRCh38/hg38 reference genome assembly was performed with NextGenMap[80]. Alignments were filtered for those with a minimum identity of 95% and minimum of 50% of the read bases mapped. For multi-mappers, ambiguous reads and non-3' UTR alignments were discarded, while one read was randomly selected from multimappers aligned to the same 3' UTR. SNP calling (coverage cut-off of 10X and variant fraction cut-off of 0.8) with VarScan2[81] was performed in order to mask actual T > C SNPs. Non-SNP T > C conversion events were then counted and the fraction of labeled transcripts determined. All results were exported (i.e., tcount file) and used for downstream analyses, such as differential expression and nascent transcript analysis.

**Bioinformatics analysis of SLAM-seq data**. The UCSC Genome Browser was used to convert RefSeq IDs to gene symbols (refGene). The resulting data were first filtered by Log2FC < 0 and sorted by Padj from lowest to highest (<0.05). The top 100 differentially-expressed genes in BC-1 and BCBL-1 were illustrated in a Venn diagram. In addition, the top 100 genes in publicly available SLAM-seq data sets derived from JQ1-treated cells[9] filtered in the same manner, were also depicted in Venn diagrams. In addition, the filtered data of BC-1 cells was analyzed using GSEA[82,83] with default parameters and adjustment of Min size = 5. For transcription factor analysis, the RefSeq IDs filtered by Log2FC < 0 and Padj < 0.05 were submitted to CSCAN[50] to analyze common regulators and to predict transcription factor binding. The following parameters were used; Organism (annotation): Homo sapiens (RefSeq), region: −450/+50, and Cell Line: GM12878. The outputs of BC-1 and BCBL-1 were then filtered with Benjamini-Hochberg adjusted p-value (Pbenj) < 0.05 and plotted in a Venn diagram.

**Purification of recombinant protein**. *Spodoptera frugiperda* Sf9 cells (Millipore) were maintained in Ex-Cell 420 medium (Sigma), and recombinant baculoviruses were generated with the BAC-to-Bac system as previously described[60,84]. The transfer plasmid, pFAST-BAC1 vectors encoding Flag-BRG1/SMARCA4, Flag-INI1/SMARCB1, and Flag-BAP170/SMARCC2 were the kind gifts from Dr. Robert Kingston (Addgene #1957, 1955, and 1953). pFAST-BAC1 encoding BAF155/ SMARCC1 and BAF60b/SMARCD2 were prepared by amplifying cDNA from pBS BAF60b and pBS BAF155 as templates and cloning also introduced a 6xHis tag sequence at the C-terminus. The pBS BAF60b and pBS BAF155 plasmids were the kind gifts from Dr. Jerry Crabtree (Addgene #17876 and 21035). The BAF57/ SMARCE1 cDNA was synthesized, which included a C-teminus 6xHis tag and cloned into the pFAST-BAC vector. Recombinant baculovirus bacmid DNA was transfected into Sf9 cells by using polyethylenimine (Sigma), and recombinant viruses were subsequently amplified once. Expression of recombinant proteins was confirmed by immunoblotting with anti-Flag monoclonal antibody (Sigma) or His tag antibody (BioRad, Hercules CA). Large-scale cultures of Sf9 cells (50 ml) were infected with recombinant baculovirus at a multiplicity of infection (MOI) of 0.1–1.0, and cells were harvested 48 h after infection. Recombinant proteins were purified after lysing infected Sf9 cell with high slat lysis buffer (Tris-HCl (pH7.5), 500 mM NaCl, 1% Triton X-100, 5% glycerol, and protease inhibitor cocktail). Cell lysates were cleared by centrifuge (7,000 rpm x 15 min at 4 °C) and incubated with either Flag-agarose beads or nickel-beads (ThermoFisher). For His-tagged protein purification, we included 10 mM immidazole and 1 mM 2-Mercaptoethanol in high salt lysis buffer. With the exception of BAF60b-His, all other proteins were successfully purified individually, and the SWI/SNF complex was prepared by mixing equal molar amounts in the presence of 1 M NaCl and then dialyzed in PBS. Complexes with three or five protein components were prepared for in vitro interaction studies. The purity and amount of protein were measured by SDS-PAGE and coomassie blue staining using bovine serum albumin (BSA) as a standard. Five protein complexes, omitting BAF60b/SMARCD2, were used for sucrose gradient sedimentation (0–50%, 35,000 rpm × 16 h with SW55 Ti rotor) in the presence or absence of VGN50 peptide (24 μM) and fractions taken from the top to the bottom of the centrifugation tube were subjected to immunoblotting with anti-Flag antibody.

**In vitro interaction assays**. Purified full-length SWI/SNF complexes containing three proteins (SMARCA4 [BRG1], SMARCC2 [BAF170], and SMARCB1 [INI1]) were mixed with GST-K-Rta deletion protein bound on glutathione beads in binding buffer (20 mM HEPES [pH 7.9], 150 mM NaCl, 1 mM EDTA, 4 mM $MgCl_2$, 1 mM dithiothreitol, 0.02% NP-40, 10% glycerol) supplemented with 1 mg/ ml BSA and 1× protease inhibitor cocktail for 30 min at 4 °C. The GST-beads were washed three times with binding buffer and subjected to SDS-PAGE after eluting proteins in sample buffer. The interaction was probed by immunoblotting with anti-Flag (Cell Signaling Technology) antibody.

**Binding ability between VGN50 and target proteins**. To evaluate of binding ability between VGN50 and each target protein, enzyme linked immunosorbent assays (ELISA) were carried out. Mutant peptide was used for comparison. Peptides were tested at the following concentrations: 0.08, 0.4. 2.0, 10, 50 nM and vehicle only (0 nM). Each protein was diluted with phosphate buffered saline (PBS) (pH = 7.0) to a final concentration of 100 nM for SMARCA4, SMARCB1, SMARCC1, SMARCC2, and SMARCE1. Each well of a 96-well flat-bottom microtiter plate was coated overnight at 4 °C with 100 μL protein in PBS. The wells were washed three times with Tris-buffered saline containing 0.1 % Tween-20 (TBS-T) with 0.1% bovine serum albumin (BSA). Blocking buffer (5% BSA in TBS-T) was added, and the plate was incubated at 37 °C for 1 h. After washing the wells three times as described above, 200 μL of different concentration of biotinylated VGN50 and mutant peptide diluted with 0.1% BSA in TBS-T were applied to each well. The plate was incubated at room temperature for 2 h and washed 5 times with TBS-T. Bound peptides were probed with 200 μL of streptavidin-horseradish peroxidase (HRP) conjugate (ThermoFisher Scientific, Waltham, MA, USA), which was diluted 1:20,000 in 0.1% BSA in TBS-T and applied to each well. After incubation at room temperature for 1.5 h and washing 5 times with TBS-T, color development with the TMB Substrate (ThermoFisher Scientific, Waltham, MA,

USA) was performed according to the manufacturer's protocol. The optical densities (ODs) were measured at 450 nm with a Benchmark Plus Microplate Spectrophotometer (Bio-Rad, Hercules, CA, USA). An OD value above background +3 SD was considered capable of binding. The assays were performed in duplicate wells and performed three times with independently-prepared recombinant proteins. Each absorbance was calculated by subtracting absorbance of the blank from the measured value in each experimental well.

**Statistics and reproducibility**. Results are shown as mean ± SD from at least three independent experiments. Data were analyzed using two-sided unpaired Student's $t$ test, ANOVA followed by Tukey's HSD test, or two-way ANOVA followed by Sidak's multiple comparison test using GraphPad Prism software (GraphPad Software, Inc., La Jolla, CA, USA). A value of $p < 0.05$ was considered statistically significant. For animal experiments, sample size is based on exploratory experiments, and were replicated twice.

**Reporting summary**. Further information on research design is available in the Nature Research Reporting Summary linked to this article.

## Data availability

Data can be accessed via the supplementary data files. Newly generated plasmids (pFAST-BAC1 SMARCE1-His and pFAST-BAC1 SMARCC1-His) are deposited in the Addgene with ID numbers, 177861 and 177863, respectively. The sequence datasets discussed in this publication have been deposited in the NCBI Gene Expression Omnibus (GEO) database as a unified Super Series (GSE173724). Source data for all plots in main figure is provided Supplementary Data 7, and uncropped blot/gel images can be found Supplementary Figure 9.

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

## Acknowledgements

We would like to thank Dr. Daniel P. Vang and Dr. Kazushi Nakano for technical support. This research was supported by public health grants from the National Cancer Institute (CA225266, CA232845), the National Institute of Dental and Craniofacial (DE025985), and the National Institute of Allergy and Infectious Diseases (AI147207, AI155515) to Y.I. The Genomics Shared Resource and the Flow Cytometry Shared Resource are supported by the UC Davis Comprehensive Cancer Center Support Grant (CCSG) awarded by the National Cancer Institute (NCI P30CA093373). M.S. was supported by the National Cancer Institute/CRCHD/CURE program.

## Author contributions

M.S. and Y.I. designed the experiments. M.S. and K.H.W. performed xenograft, flow cytometry analyses, and immunology-related experiments. Y.L., A.K., C.C., R.R.D. and Y.I. performed CUT&RUN, bioinformatics, statistical analyses, and visualization of the genomics datasets. R.R.D. and C.G.T. prepared sequencing libraries and performed initial bioinformatics analyses. C.I. and Y.I. prepared recombinant SWI/SNF proteins. H.M. and K.I.N performed ELISA and IFA and analyzed the data. Y.I. prepared SLAM-seq samples. J.F.M., J.T. and G.B. contributed reagents and provided valuable suggestions. M.S. and Y.I. wrote the manuscript and all authors edited the first and subsequent drafts.

## Competing interests

The authors declare the following competing interests: M.S. and Y.I. filed PCT Patent Application No. PCT/US2021/055979 through University of California Davis, and are founders of VGN Bio, Inc. The remaining authors declare no competing interests.
