## [Transparent Peer Review File · Communications Biology]

Reviewers' comments:

Reviewer #1 (Remarks to the Author):

Shimoda et al. present evidence that a new peptide derived from the KSHV transactivator could be useful to target MYC in cancer.

The subject is interesting to the field, but the manuscript presents several weaknesses that prevent its publication in *Communications Biology* in its current state.

General comments: The text needs to be revised, because there are several typos and errors that need careful revision. Additionally, the rationale behind each experiment and the experiments themselves are not well explained, making the read difficult and requires a lot of assumptions on the readers' part.

Major comments:

1. Caution is needed when discussing the specificity of the peptide for MYC, since it is clear that MYC is not the only gene affected by it and the peptide really shows very little regulatory overlap with JQ1, the only indirect MYC inhibitor used for comparison.
2. Explain the various cell lines used and why they were chosen.
3. Sometimes the controls used in experiments are the mutants and at other times, just the untreated. Please be consistent and use both.

Other comments:

-Inaccuracy: Omomyc is described as a 85 aminoacid protein. The most recent publications on the subject (i.e. Massó-Vallés et al, 2020) describe it instead as a 91 aminoacid protein.

-page 7: when describing the various mutants included in Figure 1, the authors should use the same nomenclature as in the figure, such as Mut1 and Mut2, because otherwise it is difficult to make the connection among them.

- Line 44-47: please, rephrase. It is not clear what the authors mean by "with different target gene profiles of BET bromodomain inhibitors, JQ1".

- Line 70: "but effectively prevent the heterodimers to bind...". Which heterodimers do the authors refer to? Myc/Max or Myc/Omomyc?

- Line 94: "two human lympho-proliferative diseases": it might be worth specifying them.

- Line 116-117: "blocks the coactivator complex". The authors have described such coactivator only in the abstract. Consider repeating the information here, please.

- The rationale for the experiments performed in the results section 'KSHV interacts with coactivators at transactivation domain' is not at all clear:

o Line 158: To rule out direct interactions, we first purified components.....: we assume that the point of this experiment is that it was performed in vitro with purified components instead of the usual incubation with cell lysates. Please make that clear.

o Line 160: What was the rationale behind each of these mutants?

o Line 165-166: "while mutation of residues 612-613 also impaired the interaction slightly with the purified proteins." The authors refer to a previous publication instead of their own data. On the contrary, their own data (Fig. 1e) shows that this mutant does not affect the interaction at all. In fact, when added to the ILQ mutation (mutant 3), it seems to rescue the lack of interaction caused by the ILQ mutation alone. Therefore, please correct the text and elaborate on why this rescue might be observed despite its contradiction with existing published data.

o In figure 1e there are some unidentified bands in the WB. Could the authors comment on those,

please? Are those isoforms, other SWI/SNF members?

- Line 170-172: "we next synthesised..... intracellular delivery". Please include a schematic in the corresponding Figure.

- Line 178-179: "which we can trigger.....inducible manner": this sentence does not make sense. Please rephrase.

-Line 179: the authors state that "(the) presence of K-RTA peptide but not mutant peptide attenuated K-Rta-mediated transactivation (Fig2a)". What is the attenuation related to? The figure shows only "no Dox" vs. "Dox" (mut or wt) and all the red samples are supposed to be upregulated compared to "no Dox", right? Please, clarify.

- Line 179-180: "as expected..... Attenuated K-Rta-mediated transcription": Which mutant peptide was used for this experiment and all the subsequent ones where the mutant peptide is used as a control? Please state it in the text.

Figure 2b: In the corresponding text, the authors state that BC-3 cells do not show reactivation. Based on the Figure, it seems that not only is reactivation not triggered in this cell line, it is in fact reduced. Can the authors explain the contradictory behaviour of this cell line?

-the mutants used in Figure 2d are different from the previous ones used in Figure 1, but there is no explanation for this switch in the text.

- Figure 2a: one of the genes in no dox is coloured in dark red when it should be white, if this is the control. Please correct.

- Figure 2a: Please include a colour scale.

- Figure 2c: Explain in accompanying text why the proteins LANA, ORF57 and K-bZIP were investigated.

- Figure 2e: Please include statistical analysis and a p-value or indicate with asterisks which differences are statistically significant.

- Line 180: Could the authors specify which genes do they refer to when they mention latent gene cluster, please? It will help the reader.

- Line 185-186: "deletion of leucine at position 610". What is the rationale behind this new mutant and why was it specifically used for this experiment in particular?

- Line 678: "we designated peptide 1 as VGN50". What about the other peptides? Were they designated with different names? This is the first time the authors define a "name" for the peptide, which does not appear in the main text. Could the authors state the name of the different peptides in a clearer way and be consistent when referring to them both in the main text and in the figures, please?

- Line 205: "as well as MYC itself (Fig. 3c)": Please clarify in the text here that the corresponding data regarding MYC was observed/validated through RT-PCR and not GSEA, as the sentence suggests.

- Line 225-226: However, we noted..... between BC-1 and BCBL-1": Please show the corresponding data.

- Line 228: the authors claim there was little gene expression altered by incubation with the mutant

peptide, but they do not show the gene expression without peptide treatment to conclude this. Can they provide these data, please?

- Figure 3d: Which cell line does this data correspond to? Please indicate in the Figure and the legend.

- Line 245: "16 down-regulated genes". In figure 3f the authors show 13 down-regulated genes in common between BC1 and BCBL-1, not 16.

- Line 256: the authors mention H3K4me3 in the text, however, in figure 4a they show PORLA and not mention it in the main text, while not showing H3K4me3 at all. Is there any reason why the SWI/SNF member picked are different from those analysed in the pull-down experiments with the peptides in figure 1? Please, comment. Also, please correct PORLA for POLRA2 if it is the gene the authors refer to.

- Figure 4b, please correct PORLA2 for POLRA2A or the appropriate gene name.

-in vivo studies: the weight

gain due to ascites is the only measure of tumor burden. Since that's not a direct measure of tumor cells at all, we would recommend using an additional quantification method, such as fluorescent imaging with luciferase expressing cells, for example.

- have the inflammatory cytokines data shown in Figure 5c been validated with any other method?

- Supplementary Fig3b: the normal ranges of the biochemistry values are missing.

Figure 6b and c, the authors only mention these results in the discussion. It is probably worth commenting on the results in the main results text.

- Mutants of antiCD22-VGN50-NLS have been used as controls in Figure 6. It would be important to show that they penetrate cells as well as the wt construct, otherwise they are not a valid control.

- What do the authors mean with the concept of evolutionarily-shaped viral proteins?

Reviewer #2 (Remarks to the Author):

The authors present intriguing data that a KSHV protein hijacks a "transcriptional activator complex", which can be used to drive leukaemia. They go on to show that a peptide derived from this viral protein can selectively kill cancer cells. Although, this is an interesting finding, there are many important unanswered questions. It is not clear to what extent the peptide blocks both MYC expression and MYC target gene activation - which is crucial to for the development of a potential therapeutic approach. A reagent that simply halt MYC expression can be circumvented by other pathways that can activate MYC. Instead, a complex that MYC itself requires for gene transcriptional activation would be a very attractive target. To push this finding forward it is really essential that the authors identify the molecular target. The ability of the viral protein to bind to the BAF complex was already shown by several other groups, and the authors have not provided any further details on this interaction. Overall the paper does not have enough substance to warrant publication at this stage. It is still too preliminary. We really need to know what cellular proteins the peptide compete with to be able to move this area forward.

Reviewer #3 (Remarks to the Author):

The paper entitled "Targeting MYC Transcription with a Small Peptide Derived from KSHV Transactivator" from Michiko Shimoda and colleagues is showing how evolutionarily-shaped viral protein sequences modulate MYC transcription and transcriptional activity. The authors demonstrated that K-Rta is inducing the transcription of viral genome by recruiting on viral genome the host transcriptional machinery composed of mediator complex, SWI/SNF components, p300 and RNA polII among others. Moreover, they synthesize a functional peptide from K-Rta (VGN50) which, by functioning as a decoy, reduce the transactivation activity of viral K-Rta, decrease the transcription of lytic viral genes and decrease cell proliferation of KSHV-positive cells. Surprisingly, the peptide is also decreasing cell growth of KSHV-negative cells, with limited effect on normal PBMC. The authors then address the transcriptional changes induced by VGN50 and identified Myc as a key downregulated gene. To address the primary transcriptional response, they took advantage of SLAMseq and they could demonstrate that VGN50 is mainly causing gene downregulation with very few upregulated genes. Since Myc was one of the top downregulated genes, they decided to compare the transcriptional profile of VGN50- and JQ1-treated cells. The number of common genes was very little, suggesting a more selective transcriptional response after VGN50 treatment. By ChIPqPCR experiments the authors demonstrated that treatment with VGN50 is decreasing the recruitment of transcriptional machinery on Myc promoter and enhancers, explaining the decrease in its mRNA levels. To demonstrate the efficacy of VGN50 in blocking cancer progression, they took advantage of a PEL xenograft model and they treated the mice with VGN50. These experiments demonstrated that VGN50 is reducing tumor growth, assessed by ascites volumes and it is changing the immune-response since a different set of cytokines could be detected in the ascites. Finally, they produced a chimeric antibody by fusing the peptide to an anti-CD22 antibody with the aim to specifically target CD22+ cells. with this experiment they demonstrated the possibility of successfully deliver VGN50 to a target cells by using specific antibodies.

The paper is very well written and rational and experiments are very well presented. The data and the approach used are very interesting and innovative and could represent a novel opportunity to target cancer cells.

Nonetheless the reviewer has few comments/revisions for the authors.

1. The title is overstating the effect on MYC transcriptional program. In the manuscript correlative data are presented with no functional validation.
2. RIME experiments should be validated with coIP experiments using KSHVpositive cells mimicking the latent and the lytic phase to prove that indeed the proteins identified with RIME are enriched in protein complex with K-Rta after viral activation
3. In the manuscript, three KSHVpositive cell lines are used: BCBL-1, BC1 and BC3. Please provide info about the cell lines in the text, highlighting the cellular differences between the three and speculating an explanation why there is no latent genes activation in BC3 after treatment with the peptide. Furthermore, no technical info regarding BC3 are present in the Material and Methods: please add.
4. In the manuscript the authors focused on the downregulation of Myc expression as well as its transcriptional program. It would be interesting to compare VGN50 treated cells with cells treated with a more specific Myc inhibitor as OmoMyc: is the transcriptional profile similar? How much is the VGN50 cellular effect actually mediated by Myc?
5. SLAMseq experiments in BC1 and BCBL-1 cells treated with VGN50 show very little overlap between the differential expressed genes. Please explain this point in more details. Please run GO or GSEA analyses to test if the pathways deregulated are similar, but differentially expressed genes reaching statistical threshold are different.
6. In line 247 the authors state that "...KSHV transactivator may preferentially take the coactivator complex from highly active (abundant) host genomic regions for their own gene expression..." there are no evidences in the manuscript for this statement: please provide data where you could demonstrate that genes downregulated after viral activation are the ones with the highest expression levels in steady state.
7. Please harmonize the text: e.g. "IP" in line 275 and "i.p." in line 279, or "S-Fig" in line 277 and

"Fig. S1" in line 141

8. Line 596: "(c) MYC..." should be in bold

9. In figure legend 5 labels are wrongly assigned and do not match with the main text or the figure. Please correct.

In general, the reviewer is positive towards the publication of the manuscript in Communications Biology journal after the points commented above will be addressed.

Response to reviewers:

Referee #1: MYC, mouse models, cancer

Referee #2: chemistry, biology, protein-protein interactions, transcription factors, drug discovery

Referee #3: epigenetics, stem cells, cancer

First of all, we would like to thank all reviewers for their time and effort for reading our manuscript and in providing us with valuable comments. Your comments enable us to revise our manuscript significantly by completing the suggested experiments to include new data sets and in correcting the text throughout. Our manuscript was also read by native speakers and copy editing by our coauthors. We listed the reviewers' comments with *Times New Roman* font and our responses were written in Arial font. Please note that due to data being reorganized to focus on the target identification of VGN50, several figure numbers are changed.

Reviewer #1 (Remarks to the Author):

Shimoda et al. present evidence that a new peptide derived from the KSHV transactivator could be useful to target MYC in cancer.

The subject is interesting to the field, but the manuscript presents several weaknesses that prevent its publication in *Communications Biology* in its current state.

General comments: The text needs to be revised, because there are several typos and errors that need careful revision. Additionally, the rationale behind each experiment and the experiments themselves are not well explained, making the read difficult and requires a lot of assumptions on the readers' part.

Thank you for these valuable comments. We revised the text extensively and include a brief introduction in the Results section to explain the rationale for each experiment. We also asked a native English speaker to review the manuscript and correct our English.

Major comments:

1. Caution is needed when discussing the specificity of the peptide for MYC, since it is clear that MYC is not the only gene affected by it and the peptide really shows very little regulatory overlap with JQ1, the only indirect MYC inhibitor used for comparison.

Thank you, and we agree with this comment. We revised the text to make it clear that the peptide is targeting a coactivator complex, which is required for MYC expression and MYC's transactivation function in PEL cells. As suggested by the reviewers, we also included Gene Set Enrichment Analyses for the SLAM-seq results, and found that MYC target genes were significantly enriched (NES: 2.90). The result is presented in Fig. 4f. Regarding JQ-1, it is a general transcription inhibitor and its target includes c-MYC. Our peptide has narrower target specificity, based on our transcriptome analyses.

	GS	NES	NOM p-val	FDR q-val
1	HALLMARK_MYC_TARGET_V1	2.90	0.000	0.000
2	HALLMARK_PI3K_AKT_MTOR_SIGNALING	1.79	0.000	0.055
3	HALLMARK_UNFOLDED_PROTEIN_RESPONSE	1.73	0.029	0.046
4	HALLMARK_MTORC1_SIGNALING	1.54	0.000	0.089

2. Explain the various cell lines used and why they were chosen.

We revised the text accordingly. We selected as many leukemia and lymphoma cell lines as possible, and one additional lung adenocarcinoma cell line that was available in our laboratory. Leukemia and lymphoma cell lines were selected since the peptide effectively killed primary effusion lymphoma cells and we wanted to reveal an association with endogenous KSHV infection.

3. Sometimes the controls used in experiments are the mutants and at other times, just the untreated. Please be consistent and use both.

We could not identify experiments that only non-treated sample was used as a control. Nonetheless, we went through all figures and included appropriate controls in figures.

Other comments:

1. Inaccuracy: Omomyc is described as an 85 amino acid protein. The most recent publications on the subject (i.e. Massó-Vallés et al, 2020) describe it instead as a 91 amino acid protein.

We appreciate your comments. We found that the information is mixed regarding the length of Omomyc. Based on the information on page 9 of Cells 2020, 9, 883 by Massó-Vallés et al, 2020 (PMID 32260326) we corrected the text as a “90 amino acid protein”.

2. page 7: when describing the various mutants included in Figure 1, the authors should use the same nomenclature as in the figure, such as Mut1 and Mut2, because otherwise it is difficult to make the connection among them.

We appreciate your comments. We revised the text on page 7 accordingly to specify which parts correspond to the results obtained using Mut1, 2, and 3.

3. Line 44-47: please, rephrase. It is not clear what the authors mean by “with different target gene profiles of BET bromodomain inhibitors, JQ1”.

We revised the text as shown below.

“The top 100 down-regulated genes were extracted and examined for similarity among down-regulated genes. The results showed that very few cellular genes were commonly down-regulated between the BET bromodomain inhibitor JQ1 and VGN50 “

4. Line 70: ”but effectively prevent the heterodimers to bind...”. Which heterodimers do the authors refer to? Myc/Max or Myc/Omomyc?

We specified this as Myc/Max in the text.

5. Line 94: “two human lympho-proliferative diseases”: it might be worth specifying them.

We did not change the text since these two diseases were previously specified in the text as “two human lympho-proliferative diseases, primary effusion lymphoma (PEL) (19,

20) and multicentric Castleman's disease (21, 22), “

6. Line 116-117: “blocks the coactivator complex”. The authors have described such coactivator only in the abstract. Consider repeating the information here, please.

We inserted the following description to provide specific information.

“..blocks the coactivator complexes consisting of Nuclear receptor coactivator 2, p300, and SWI/SNF proteins from engaging the *MYC* promoter”

7. The rationale for the experiments performed in the results section ‘KSHV interacts with coactivators at transactivation domain’ is not at all clear:

Line 158: To rule out direct interactions, we first purified components.....: we assume that the point of this experiment is that it was performed in vitro with purified components instead of the usual incubation with cell lysates. Please make that clear.

We revised the section as below to make the context clear.

“To rule out indirect interactions, which may result from crude cell lysates, we purified components of the SWI/SNF complex from recombinant baculovirus-infected Sf9 cells. The K-Rta transactivation domain (Fig. 1e) and its mutant were expressed as GST-fusion proteins and purified from *E.coli* (Mut1, 2, and 3 in Fig. 1f, g). The GST-pull down studies using wild type (WT) K-Rta domain showed that the SWI/SNF complex containing SMARCA4, SMARCC2, and SMARCB1, directly interacted with K-Rta between residues 551 and 650. “

8. Line 160: What was the rationale behind each of these mutants?

We revised the section as shown below to make the context clear.

We targeted amino acids residues that were conserved among other gamma-herpesviral homologous proteins. To make it clearer, we also included an alignment of the protein sequences in Fig. 2a.

9. Line 165-166: “while mutation of residues 612-613 also impaired the interaction slightly with the purified proteins.” The authors refer to a previous publication instead of their own data. On the contrary, their own data (Fig. 1e) shows that this mutant does not affect the interaction at all. In fact, when added to the ILQ mutation (mutant 3), it seems to rescue the lack of interaction caused by the ILQ mutation alone. Therefore, please correct the text and elaborate on why this rescue might be observed despite its contradiction with existing published data.

Thank you. We included a possible explanation as follows:

Additional mutation of negatively charged amino acids (DD) to hydrophobic amino acids (AA) in Mut 3 slightly restored interaction of Mut2 with SMARCB1. The Mut2 has a mutation reduces hydrophobicity due to alanine substitution (ILQ/AAA). The result may indicate that the hydrophobicity of the sequence stretch plays an important role in the interaction between K-Rta and SMARCB1.

10. In figure 1e there are some unidentified bands in the WB. Could the authors comment on those, please? Are those isoforms, other SWI/SNF members?

This is a very good observation and point. Currently, we do not know the identity of the proteins with MW ~100kDa. We assume that it might be degradation products of SMARCA4 or SMARCC2 that still contain Flag tag.

11. Line 170-172: “ we next synthesized..... intracellular delivery”. Please include a schematic in the corresponding Figure.

We revised the corresponding section of the text and also prepared a new Fig. 2a and b that includes a schematic to make the content clearer.

12. Line 178-179: “which we can trigger.....inducible manner”: this sentence does not make sense. Please rephrase.

We revised the sentence as shown below to make it clearer.

“To study this, we used the TREx-K-Rta BCBL-1 cell line, in which we can trigger the expression of exogenous Flag-tagged K-Rta in a doxycycline inducible manner.”

13. Line 179: the authors state that “(the) presence of K-RTA peptide but not mutant peptide attenuated K-Rta-mediated transactivation (Fig2a)”. What is the attenuation related to? The figure shows only “no Dox” vs. “Dox” (mut or wt) and all the red samples are supposed to be upregulated compared to “no Dox”, right? Please, clarify.

- Line 179-180: “as expected..... Attenuated K-Rta-mediated transcription”: Which mutant peptide was used for this experiment and all the subsequent ones where the mutant peptide is used as a control? Please state it in the text.

We revised the text accordingly and also prepared a new version of Fig. 3a to make contents clearer (Page 11 line 2-8).

14. Figure 2b: In the corresponding text, the authors state that BC-3 cells do not show reactivation. Based on the Figure, it seems that not only is reactivation not triggered in this cell line, it is in fact reduced. Can the authors explain the contradictory behavior of this cell line?

This is a very important question. We still do not know what makes this cell line respond differently. BC3, but not BC1 or BCBL-1, contain TP53 mutation, which may be associated with the phenotypic differences. To avoid confusion, we omitted the panels of BC3 from all figures, but plan to further investigate this in the future with transcriptome analyses. With MTT assays, BC-3 cells are also approximately 4-fold more resistant to VGN50 than BC1 or BCBL-1.

15. the mutants used in Figure 2d are different from the previous ones used in Figure 1, but there is no explanation for this switch in the text.

We have made this clearer by including sequence alignment figure and peptide sequences in Fig. 2a, b.

16. Figure 2a: one of the genes in no dox is colored in dark red when it should be white, if this is the control. Please correct.

Thank you. We have corrected this.

17. Figure 2a: Please include a colour scale.

We now included a color scale.

18. Figure 2c: Explain in accompanying text why the proteins LANA, ORF57 and K-bZIP were investigated.

We expanded the text to include the rationale for investigating these proteins.

19. Figure 2e: Please include statistical analysis and a p-value or indicate with asterisks which differences are statistically significant.

We included statistical analysis and the corresponding p-values to show statistically significant differences.

20. Line 180: Could the authors specify which genes do they refer to when they mention latent gene cluster, please? It will help the reader.

We deemphasize the effects on KSHV replication in order to focus more on cellular effects. We therefore omitted the sentence.

21. Line 185-186: “deletion of leucine at position 616”. What is the rationale behind this new mutant and why was it specifically used for this experiment in particular?

We included an alignment with other gamma-herpesviral proteins and expanded text to explain the rationale.

22. Line 678: “we designated peptide 1 as VGN50”. What about the other peptides? Were they designated with different names? This is the first time the authors define a “name” for the peptide, which does not appear in the main text. Could the authors state the name of the different peptides in a clearer way and be consistent when referring to them both in the main text and in the figures, please?

We expanded a figure 2 to explain better. We only selected the shortest and the most potent peptide for cell killing. We did not name other peptides that were used for selection process.

23. Line 205: “as well as MYC itself (Fig. 3c)”: Please clarify in the text here that the corresponding data regarding MYC was observed/validated through RT-PCR and not GSEA, as the sentence suggests.

We revised the section as below.

“GSEA also showed that MYC target genes, which include enzymes associated with DNA replication, were among the significantly down-regulated gene sets (Fig. 4a, b). Furthermore, we confirmed the result by RT-qPCR that the expression of MYC itself was decreased in both BCBL-1 and BC-1 cells after treatment with VGN50, but not mutant peptide (Fig.4c).

” (please note that due to new data added, figure numbers are changed)

24. Line 225-226: However, we noted..... between BC-1 and BCBL-1”: Please show the corresponding data.

Line 228: the authors claim there was little gene expression altered by incubation with the mutant peptide, but they do not show the gene expression without peptide treatment to conclude this. Can they provide these data, please?

We are sorry for confusion. The waterfall figure for the mutant is demonstrating differentially regulated transcripts between mock-treated and mutant peptide treated. Similarly, figure presented for wild type is showing transcripts differentially expressed by VGN50 treatment. Accordingly, both samples are comparing with same sets of SLAM-seq data (Mock-treated) so that we can compare degree of gene expression changes. We expanded text to make it clearer.

25. Figure 3d: Which cell line does this data correspond to? Please indicate in the Figure and the legend.

Thank you. Figure 3d is now Fig. 4d and BC-1 is labeled in it.

26. Line 245: “16 down-regulated genes”. In figure 3f the authors show 13 down-regulated genes in common between BC1 and BCBL-1, not 16.

Thank you, and we apologize for any confusion by the Venn diagram. There are 16 genes in common between VGN50-treated BC1 and BCBL-1 that are divided into subsets of 13 and 3 genes due to the latter 3 genes also being in common with JQ-1.

27. Line 256: the authors mention H3K4me3 in the text, however, in figure 4a they show PORLA and not mention it in the main text, while not showing H3K4me3 at all. Is there any reason why the SWI/SNF member picked are different from those analysed in the pull-down experiments with the peptides in figure 1? Please, comment. Also, please correct PORLA for POLRA2 if it is the gene the authors refer to.

We apologize for the confusion. We meant H3K4me1, which is a marker for enhancers. Both SMARCC1 and SMARCA4 are proteins identified in RIME studies. We also included peptide ELISA studies in this revised manuscript to show that they interact with VGN50. The

differences in the antibody used are due to efficacies of the antibody for CUT&RUN experiments. We later found that the SMARCA4 antibody from Epicypher works well in that it had less background noise in CUT&RUN studies, so we decided to use this reagent to characterize SWI/SNF complex recruitment. By using the two antibodies targeting the same protein complex, we concluded that SWI/SNF complex recruitment was inhibited in the presence of VGN50. We also corrected the typographical error to POLR2A. Thank you.

28. Figure 4b, please correct PORLA2 for POLRA2A or the appropriate gene name.

We corrected to POLR2A. Thank you.

29. in vivo studies: the weight gain due to ascites is the only measure of tumor burden. Since that's not a direct measure of tumor cells at all, we would recommend using an additional quantification method, such as fluorescent imaging with luciferase expressing cells, for example.

We appreciate the reviewer's suggestions and regret that we did not use Luciferase expressing cell lines in this experiment. We used KSHV-associated aggressive PEL cell lines that grow in the peritoneal cavity and accumulate ascites. However, to validate PEL growth, in addition to the ascites volume and cell count (Fig.6b), we conducted flow cytometric analysis of the PEL cells from the ascites to confirm that PEL growth in the peritoneal cavity of the VGN50-peptide treated group was significantly reduced in association with atrophy compared to that of the controls (PBS and mutant peptide treatment) groups (Fig. 6C). These are explained in the main text as follows.

"Based on the drug schedule, we examined the effects of the peptide on PEL xenograft cell growth. In this established model, BCBL-1 cells grow in the peritoneal cavity resulting in accumulation of ascites, which can be measured as a body weight gain that corresponds to the volume of ascites (56). We inoculated male NRG mice with 2×10^7 BCBL-1 cells via i.p. injection followed by i.p. injections of VGN50 (10 mg/kg), mutant control peptide or PBS, daily for 10 days beginning two days after (day 0) tumor inoculation. At the termination point, tumor growth was also validated by flow cytometric analysis of the PEL cells in the ascites fluid (Fig. 6b and c).

30. have the inflammatory cytokines data shown in Figure 5c been validated with any other method?

Regrettably, we did not validate the cytokine levels with another method.

31. Supplementary Fig3b: the normal ranges of the biochemistry values are missing.

The normal ranges for these values are now added to the S-Fig.4b.

32. Figure 6b and c, the authors only mention these results in the discussion. It is probably worth commenting on the results in the main results text.

- Mutants of antiCD22-VGN50-NLS have been used as controls in Figure 6. It would be important to show that they penetrate cells as well as the wt construct, otherwise they are not a valid control.

This is an excellent point. We extended these studies with antiCD22-VGN50 constructs, and recently found that the recombinant protein may also function at the cell surface by disrupting receptor ligand interactions. Accordingly, we decided to clarify the molecular mechanisms further, which may also result in alternative approach to using the VGN50 sequence. By knowing that the recombinant protein works by a distinct molecular mechanism, we decided to omit previous Figure 6 to avoid confusion and distract from the main point of the report.

33. What do the authors mean with the concept of evolutionarily-shaped viral proteins?

We clarify this in the text as follows:

Our study demonstrated that a viral protein is a unique starting material as the basis for designing therapeutics directed at attenuating cellular protein function(s). We hypothesize that viral proteins continuously evolved structures possessing enhanced efficiencies to hijack cellular protein function, and resulting in the current conserved amino acid sequence.

Reviewer #2 (Remarks to the Author):

The authors present intriguing data that a KSHV protein hijacks a "transcriptional activator complex", which can be used to drive leukemia. They go on to show that a peptide derived from this viral protein can selectively kill cancer cells. Although, this is an interesting finding, there are many important unanswered questions. It is not clear to what extent the peptide blocks both MYC expression and MYC target gene activation - which is crucial to for the development of a potential therapeutic approach. A reagent that simply halt MYC expression can be circumvented by other pathways that can activate MYC. Instead, a complex that MYC itself requires for gene transcriptional activation would be a very attractive target. To push this finding forward it is really essential that the authors identify the molecular target. The ability of the viral protein to bind to the BAF complex was already shown by several other groups, and the authors have not provided any further details on this interaction. Overall the paper does not have enough substance to warrant publication at this stage. It is still too preliminary. We really need to know what cellular proteins the peptide competes with to be able to move this area forward.

First of all, we would like to thank the reviewer for taking the time to provide these very helpful comments on our manuscript. The following are our responses to your critiques.

1. It is not clear to what extent the peptide blocks both MYC expression and MYC target gene activation - which is crucial to for the development of a potential therapeutic approach.

We generally agree with your comments. We also think your critique is very difficult to answer. The extent of MYC inhibition will depend on the amount of peptide with the cells are treated and when MYC expression is examined.

However, our peptide is potent enough to kill cancer cells in vitro and in vivo. We included the results of GSEA conducted on VGN50 direct targets (i.e., obtained from SLAM-seq datasets) in this revision. The results clearly showed that VGN50 can inhibit MYC target

	GS	NES	NOM p-val	FDR q-val
1	HALLMARK_MYC_TARGET_V1	2.90	0.000	0.000
2	HALLMARK_PI3K_AKT_MTOR_SIGNALING	1.79	0.000	0.055
3	HALLMARK_UNFOLDED_PROTEIN_RESPONSE	1.73	0.029	0.046
4	HALLMARK_MTORC1_SIGNALING	1.54	0.000	0.089

gene expression as evidence by the very high normalized enrichment score (NES) of 2.90 (Fig. 4f). The results suggested that, at the minimum, VGN50 preferentially targets MYC and MYC target gene expression in PEL cells. We also provide evidence that MYC gene expression is inhibited in presence of VGN50 (Fig. 4e). We also expanded discussion to address this.

2. A reagent that simply halt MYC expression can be circumvented by other pathways that can activate MYC. Instead, a complex that MYC itself requires for gene transcriptional activation would be a very attractive target.

We agreed and we would like to thank you for kind words. Additionally, we hope that our results strongly support this.

3. To push this finding forward it is really essential that the authors identify the molecular target. The ability of the viral protein to bind to the BAF complex was already shown by several other groups, and the authors have not provided any further details on this interaction.

We individually isolated SWI/SNF complex components and performed ELISA-based binding assays to confirm their interactions.

The results are included in Fig. 3d,e,f. The results demonstrated that VGN50 interacts well with SMARCC2, SMARCE1, and SMARCB1. While we cannot rule out the possibility that VGN50 interacts with other cellular molecules and their indirect effects, genomic studies showed that the SWI/SNF complex binding to MYC regulatory regions are clearly inhibited in the presence of VGN50 in tissue culture and led to cancer cell death.

Reviewer #3 (Remarks to the Author):

The paper entitled “Targeting MYC Transcription with a Small Peptide Derived from KSHV Transactivator” from Michiko Shimoda and colleagues is showing how evolutionarily-shaped viral protein sequences modulate MYC transcription and transcriptional activity. The authors demonstrated that K-Rta is inducing the transcription of viral genome by recruiting on viral genome the host transcriptional machinery composed of mediator complex, SWI/SNF components, p300 and RNA polII among others. Moreover, they synthesize a functional peptide from K-Rta (VGN50) which, by functioning as a decoy, reduce the transactivation activity of viral K-Rta, decrease the transcription of lytic viral genes and decrease cell proliferation of KSHV-positive cells. Surprisingly, the peptide is also decreasing cell growth of KSHV-negative cells, with limited effect on normal PBMC. The authors then address the transcriptional changes induced by VGN50 and identified Myc as a key downregulated gene. To address the primary transcriptional response, they took advantage of SLAMseq and they could demonstrate that VGN50 is mainly causing gene downregulation with very few upregulated genes. Since

Myc was one of the top downregulated genes, they decided to compare the transcriptional profile of VGN50- and JQ1-treated cells. The number of common genes was very little, suggesting a more selective transcriptional response after VGN50 treatment. By ChIPqPCR experiments the authors demonstrated that treatment with VGN50 is decreasing the recruitment of transcriptional machinery on Myc promoter and enhancers, explaining the decrease in its mRNA levels. To demonstrate the efficacy of VGN50 in blocking cancer progression, they took advantage of a PEL xenograft model and they treated the mice with VGN50. These experiments demonstrated that VGN50 is reducing tumor growth, assessed by ascites volumes and it is changing the immune-response since a different set of cytokines could be detected in the ascites. Finally, they produced a chimeric antibody by fusing the peptide to an anti-CD22 antibody with the aim to specifically target CD22+ cells. with this experiment, they demonstrated the possibility of successfully deliver VGN50 to a target cells by using specific antibodies.

The paper is very well written and rational and experiments are very well presented. The data and the approach used are very interesting and innovative and could represent a novel opportunity to target cancer cells. Nonetheless the reviewer has few comments/revisions for the authors.

We greatly appreciate your support and kind words.

1. The title is overstating the effect on MYC transcriptional program. In the manuscript correlative data are presented with no functional validation.

This point is very well taken. We therefore modified our title to "KSHV transactivator-derived small peptide traps coactivators to attenuate MYC function and kills leukemia and lymphoma " according to your suggestions.

2. RIME experiments should be validated with coIP experiments using KSHV positive cells mimicking the latent and the lytic phase to prove that indeed the proteins identified with RIME are enriched in protein complex with K-Rta after viral activation.

We have performed the suggested co-IP experiments and included a figure with the results that confirmed the interaction of K-Rta with NCoA2 (Fig. 1C).

3. In the manuscript, three KSHV positive cell lines are used: BCBL-1, BC1 and BC3. Please provide info about the cell lines in the text, highlighting the cellular differences between the three and speculating an explanation why there is no latent genes activation in BC3 after treatment with the peptide. Furthermore, no technical info regarding BC3 are present in the Material and Methods: please add.

Thank you. We now include a description of BC3 in the Methods. In this revised manuscript, we also decided to omit BC3 data sets from the main figures since BC3, but not BC1 or BCBL-1, contains TP53 mutations. BC3 also demonstrated a slightly different response to VGN50, which we cannot explain at this moment. BC3 cells are also approximately 4-times more

Thank you. We now include a description of BC3 in the Methods. In this revised manuscript, we also decided to omit BC3 data sets from the main figures since BC3, but not BC1 or BCBL-1, contains TP53 mutations. BC3 also demonstrated a slightly different response to VGN50, which we cannot explain at this moment. BC3 cells are also approximately 4-times more

resistant to VGN50. We will further investigate this with transcriptome analyses to define the VGN50 drug resistance mechanism in the future.

4. In the manuscript the authors focused on the downregulation of Myc expression as well as its transcriptional program. It would be interesting to compare VGN50 treated cells with cells treated with a more specific Myc inhibitor as OmoMyc: is the transcriptional profile similar? How much is the VGN50 cellular effect actually mediated by Myc?

This is a very good point and great suggestions. We would like to perform the suggested experiments side by side with transcriptomic analyses. However, our current study is focused on the identification of a peptide drug from a viral protein and its initial characterization. Comparisons with other existing drugs and combinations with other drugs for synergistic cancer cell killing effects will certainly be performed in future studies.

5. SLAMseq experiments in BC1 and BCBL-1 cells treated with VGN50 show very little overlap between the differential expressed genes. Please explain this point in more details. Please run GO or GSEA analyses to test if the pathways deregulated are similar, but differentially expressed genes reaching statistical threshold are different.

For the SLAM-seq analyses, we used a VGN50 treatment at a concentration of 24 μ M for 30 mins. BC-1 cells are more sensitive than BCBL-1 and demonstrated a higher number of down-regulated genes than BCBL-1 with the same cut-off criteria. Due to a lower number of genes, we could not obtain enough statistical power for GO and GSEA analyses for BCBL-1. However, with BC-1 cells, GSEA performed on the SLAM-seq results identified strong enrichment of MYC target genes with 2.90 NES (Fig. 4f). With CSCAN analyses, we could also identify significantly overlapping target molecules, suggesting that the molecular action of the peptide is very similar in both BC1 and BCBL-1 (Fig. 4h).

	GS	NES	NOM p-val	FDR q-val
1	HALLMARK_MYC_TARGET_V1	2.90	0.000	0.000
2	HALLMARK_PI3K_AKT_MTOR_SIGNALING	1.79	0.000	0.055
3	HALLMARK_UNFOLDED_PROTEIN_RESPONSE	1.73	0.029	0.046
4	HALLMARK_MTORC1_SIGNALING	1.54	0.000	0.089

6. In line 247 the authors state that "...KSHV transactivator may preferentially take the coactivator complex from highly active (abundant) host genomic regions for their own gene expression..." there are no evidences in the manuscript for this statement: please provide data where you could demonstrate that genes downregulated after viral activation are the ones with the highest expression levels in steady state.

Thank you. This point is well taken, and we omitted this description.

7. Please harmonize the text: e.g. "IP" in line 275 and "i.p." in line 279, or "S-Fig" in line 277 and "Fig. S1" in line 141

We harmonized the text as "i.p." throughout the manuscript.

8. Line 596: "(c) MYC..." should be in bold

9. In figure legend 5 labels are wrongly assigned and do not matched with the main text or the figure. Please correct.

Thank you for your time and careful review. We corrected the labels in the text.

Reviewers' comments:

Reviewer #1 (Remarks to the Author):

The authors have been thorough with the revision and have addressed our concerns.

Reviewer #2 (Remarks to the Author):

The authors in the revised manuscript try to identify the molecular target, in particular which of the SWI/SNF subunits the short peptide targets. Human SWI/SNF complexes include three main complex assemblies: cBAF, PBAF and ncBAF. The authors' proteomic data show that the peptide interacts with cBAF. So, to try to identify biochemically which subunit the peptide specifically targets, they set-up an ELISA-based assay where they use several isolated subunits. First of all, it is really surprising that they only focus on the subunits SMARCA4 (also called BRG1), SMARCB1 (also called INI1 or SNF5), SMARCC1 (also called BAF155), SMARCC2 (also called BAF170), and SMARCE1 (also called BAF57) as these are common to several other SWI/SNF assemblies, and they do not test the peptide against any of the characteristic subunits that are unique to the cBAF complex – which would appear to be likely candidates for this putative interaction.

The results of the ELISA study are, unfortunately, problematic. The peptide in this assay has an affinity for all of the subunits, more for some, less for others, but still seems to bind to all of them. Based on the length of the peptide and the structural information that we have recently acquired for the complex is not feasible that such a short motif can bind in any way to all of the subunits – unless we are looking at a non-specific interaction! The authors do not carry out any type of analysis to verify that the individual subunits are properly folded/stable in isolation, or functional. As we understand from the structural studies of the whole complex, each of the subunits includes regions that form coil-coiled interactions with other subunits, and thus would be particularly “sticky” outside of the context of the complex. This would explain why the peptide makes non-specific interactions. It is also interesting to note that in the text, when discussing the results of the ELISA assay (Figure 3), the authors mention that the peptide interacts well with SMARCA4, SMARCB1, and SMARCE1 – but THEN in the rebuttal letter it seems that they change their minds as they say that the biochemical assay demonstrates that the peptide preferably binds to SMARCC2, SMARCE1, and SMARCB1. To me this seems that also the authors may be a bit sceptical about these results. Unfortunately, these biochemical studies have not been able to properly address the molecular target of the peptide – quite the opposite they are worsening the quality of the overall study. It is indeed challenging to determine the molecular target of a peptide for such a large assembly, so the authors should appreciate that this requires further studies using other techniques and approaches. It really requires more effort and time.

Reviewer #3 (Remarks to the Author):

Shimoda and colleagues nicely addressed all the reviewer's comments and suggestions. The reviewer is therefore in favor of a publication of the paper entitled “KSVH transactivator-derived small peptide traps coactivators to attenuate MYC function and kills leukemia and lymphoma” in *Communications Biology*.

Response to the Reviewer's remarks

Reviewer #2 (Remarks to the Author):

The authors in the revised manuscript try to identify the molecular target, in particular which of the SWI/SNF subunits the short peptide targets. Human SWI/SNF complexes include three main complex assemblies: cBAF, PBAF and ncBAF. The authors' proteomic data show that the peptide interacts with cBAF. So, to try to identify biochemically which subunit the peptide specifically targets, they set-up an ELISA-based assay where they use several isolated subunits. First of all, it is really surprising that they only focus on the subunits SMARCA4 (also called BRG1), SMARCB1 (also called INI1 or SNF5), SMARCC1 (also called BAF155), SMARCC2 (also called BAF170), and SMARCE1 (also called BAF57) as these are common to several other SWI/SNF assemblies, and they do not test the peptide against any of the characteristic subunits that are unique to the cBAF complex – which would appear to be likely candidates for this putative interaction. The results of the ELISA study are, unfortunately, problematic. The peptide in this assay has an affinity for all of the subunits, more for some, less for others, but still seems to binds to all of them. Based on the length of the peptide and the structural information that we have recently acquired for the complex is not feasible that such a short motif can bind in any way to all of the subunits – unless we are looking at a non-specific interaction! The authors do not carry out any type of analysis to verify that the individual subunits are properly folded/stable in isolation, or functional. As we understand from the structural studies of the whole complex, each of the subunits includes regions that form coil-coiled interactions with other subunits, and thus would be particularly “sticky” outside of the context of the complex. This would explain why the peptide makes non-specific interactions. It is also interesting to note that in the text, when discussing the results of the ELISA assay (Figure 3), the authors mention that the peptide interacts well with SMARCA4, SMARCB1, and SMARCE1 – but THEN in the rebuttal letter it seems that they change their minds as they say that the biochemical assay demonstrates that the peptide preferably binds to SMARCC2, SMARCE1, and SMARCB1. To me this seems that also the authors may be a bit sceptical about these results.

Unfortunately, these biochemical studies have not been able to properly address the molecular target of the peptide – quite the opposite they are worsening the quality of the overall study. It is indeed challenging to determine the molecular target of a peptide for such a large assembly, so the authors should appreciate that this requires further studies using other techniques and approaches. It really requires more effort and time.

Thank you very much for your critical remarks.

We agree with you that the ELISA-type assay we used in this study has limitations to fully understand the mechanism by which VGN binds to SWI/SNF subunits, especially for the large SWI/SNF complexes. We also agree that the solid-phase ELISA may not faithfully recapitulate the protein-protein interaction in vivo. As you pointed out, the ELISA results could be due to the “sticky” nature of these SWI/SNF components, especially when these proteins are immobilized in the solid phase. On the other hand, the ELISA results demonstrated that VGN50, but not a control peptide lacking the ILQ motif, bound to some of the SWI/SNF components in a dose-dependent manner. Such results, at least, support the flexible binding property of VGN50 and significance of the conserved motif.

We sincerely appreciate that identifying the molecular target of VGN50 is not a simple task utilizing ELISA. It requires further studies using other appropriate techniques and approaches, and of course, more effort and time. We are very interested in the mechanism and eager to know how VGN50 can block MYC transcriptional complex assembly. We believe VGN50 can be used as a probe to better understand SWI/SNF biology and for developing new cancer therapeutics. We hope to examine the interaction between VGN50 and SWI/SNF, especially the cBAF complex as you suggested, to identify the molecular target(s) of VGN50 in our future studies.

With those in mind, we revised our manuscript as described below in two places. We hope this addressed your concerns and you agree that our manuscript to be published in Communication Biology.

Page 11-12

To confirm the basis of the VGN50 action, the biochemical interactions between VGN50 and SWI/SNF proteins (i.e., putative VGN50 target molecules) were ~~confirmed~~examined by ELISA-based binding assays using purified 5 individual SWI/SNF components prepared from baculovirus-infected Sf9 cells (**Fig. 3d**).

Page 22 (the end of the discussion)

...loss of weight (**Supplementary Fig. 4**).

Human SWI/SNF complexes are remarkably large, including the products of the 29 genes encoding mSWI/SNF subunits that assemble into three distinct mSWI/SNF complexes, termed canonical BRG1/BRM associated factor (cBAF), polybromo-associated BAF (PBAF), and noncanonical BAF (ncBAF), each of which comprises common as well as complex-specific subunits (70). Recent studies have revealed a high prevalence of mutations in genes encoding subunits of the SWI/SNF complexes in cancers and neurological diseases (70, 71), indicating the importance of this complex as a therapeutic target. In this context, our current study showed that the oncogenic KSHV utilizes the SWI/SNF complex for viral genome replication through the interaction with the viral transactivator protein K-Rta. Moreover, we showed that VGN50, a K-Rta-derived peptide, can be used to prevent the SWI/SNF complex from forming the MYC transactivation machinery in cancer cells. The limitation of the current study is that we could not pinpoint the specific molecular target and structural mechanism through which VGN50 interacts with this large SWI/SNF complex. By ELISA-based assays, we attempted to show that VGN50, but not a control mutant peptide which lacks the highly conserved ILQ sequence motif among viral transcription factors, biochemically binds to several isolated shared subunits of SWI/SNF complexes. However, further studies are required using other techniques and approaches to fully understand the structural mechanism in which this small viral-derived peptide can flexibly interact with several SWI/SNF components and capture subunits of SWI/SNF, thus preventing it from assembling a transactivation complex on the MYC promoter. The latter is especially important to better understanding SWI/SNF biology and to identify new approaches for targeting SWI/SNF complexes in cancer therapy.

In summary, our study demonstrated that....